# Aligning Collaborative View Recovery and Tensorial Subspace Learning via Latent Representation for Incomplete Multi-view Clustering

**Youqing Wang**[1]  **Yu Cao**[1]  **Jinlu Wang** [2]  **Xiang Xu** [1]  **Jiapu Wang** [3]  **Tengfei Liu**[2]
**Junbin Gao** [4]  **Jipeng Guo**[1, ✉]

[1] College of Information Science and Technology, Beijing University of Chemical Technology
[2] Beijing University of Technology, [3] Nanjing University of Science and Technology
[4] The University of Sydney Business School, The University of Sydney
{wang_youqing,2024210615,2024200756,guojipeng}@buct.edu.cn
{jinluwang,tfliu}@emails.bjut.edu.cn
jiapuwang9@gmail.com,junbin.gao@sydney.edu.au

## Abstract

Multi-view data usually suffer from partially missing views in open scenarios, which inevitably degrades clustering performance. The incomplete multi-view clustering (IMVC) has attracted increasing attention and achieved significant success. Although existing imputation-based IMVC methods perform well, they still face one crucial limitation, i.e., view recovery and subspace representation lack explicit alignment and collaborative interaction in exploring complementarity and consistency across multiple views. To this end, this study proposes a novel IMVC method to **A**lign collaborative view **R**ecovery and tensorial **S**ubspace **L**earning via latent representation (ARSL-IMVC). Specifically, the ARSL-IMVC infers the complete view from view-shared latent representation and view-specific estimator with Hilbert-Schmidt Independence Criterion regularizer, reshaping the consistent and diverse information intrinsically embedded in original multi-view data. Then, the ARSL-IMVC learns the view-shared and view-specific subspace representations from latent feature and recovered views, and models high-order correlations at the global and local levels in the unified low-rank tensor space. Thus, leveraging the latent representation as a bridge in a unified framework, the ARSL-IMVC seamlessly aligns the complementarity and consistency exploration across view recovery and subspace representation learning, negotiating with each other to promote clustering. Extensive experimental results on seven datasets demonstrate the powerful capacity of ARSL-IMVC in complex incomplete multi-view clustering tasks under various view missing scenarios. The source code is publicly available at https://github.com/caoyu110/ARSL-IMVC.

## 1 Introduction

Recently, the rapid development of information technology promotes the complexity of data forms, including text, audio, images, etc. Extracting available information from multi-view data collected from different sources is challenging, especially in unsupervised scenarios (Xu et al., 2013; Guo et al., 2025a; Zhao et al., 2026). As a key unsupervised learning technology, clustering is also expanded to multi-view case to better accommodate complex multi-view data (Chao et al., 2021; Chen et al., 2022; Lou et al., 2025). Multi-view clustering (MVC) aims to divide unlabeled multi-view data into disjoint clusters by leveraging the consistency and complementarity across multi-view data (Wang et al., 2025a;b). Among various MVC approaches, multi-view subspace clustering (Gao et al., 2015; Cao et al., 2015; Shi et al., 2024) receives considerable attention due to its significant performance advantage and robustness. It aims to learn optimal subspace representations in which the samples can be more clearly separated into distinct clusters, simultaneously preserving

the complementary information and mitigating redundancy and noise across multiple views. When both inter-view consistency and view-specific diversity are fully explored and well balanced, the clustering performance can be significantly enhanced (Guo et al., 2023).

However, in open scenarios, due to sensor failure, missing annotations, or data corruption, etc, it is usually difficult to obtain complete data for all views. Reducing the negative impact of incompleteness of multiple views on clustering performance becomes a major challenge currently faced in the MVC filed (Wen et al., 2023b). And thus, the incomplete multi-view clustering (IMVC) methods are widely proposed and divided into two categories, i.e., imputation-based and imputation-free methods (Liu et al., 2025; Wen et al., 2024b). The imputation-based IMVC methods adhere to the mechanism of first view recovery and then clustering structure exploration (Liu et al., 2024a; Wang et al., 2021). The imputation-free IMVC methods only focus on partial observable views to explore clustering information, avoiding computational costs of missing view completion (Wen et al., 2024b; Qin et al., 2025; Wen et al., 2026). Despite strong simplicity of imputation-free IMVC methods, their discriminability to extract clustering information is restricted by limited available views, especially when the missing rate is high.

The imputation-based IMVC methods utilize heuristic or learnable strategies to impute the missing views, providing strong data foundation for clustering information exploration and enhancing the interpretability. Surprisingly, several methods jointly recover the missing views and learn clustering representations (subspace coefficient, graph similarity, cluster indicator, etc) in a unified framework, significantly improving clustering quality. Despite the success, two critical challenges remain in most existing imputation-based IMVC methods. First, the recovered or completed views often suffer from limited structural fidelity and insufficient diversity and consistency reshaping, which are both essential for effective multi-view clustering. More important, there is no explicit alignment and collaborative interaction in view recovery and subspace representation learning in exploring complementarity and consistency.

To this end, this study unifies collaborative view completion and tensorial subspace learning, and breaks the gap between them in complementarity and consistency modeling by shared latent representation. Notably, the latent representation not only serves as fictitious transitional factor for view reconstruction but also directly contributes to the subspace learning with structural awareness. Along with view-specific diversity term, the proposed method provides more freedom in enriching feature description across views. Consequently, the shared and specific subspace representations derived from the latent space and imputed views are integrated into low-rank tensor, enabling interactions across different levels of structural information. In summary, the primary contributions of this study are as follows:

- The proposed novel ARSL-IMVC method facilitates the unified and explicit alignment of complementarity and consistency exploration across both missing feature reconstruction and subspace representation learning, fostering a coherent cross-view correlation modeling.

- The complex high-order correlation among local specific and global shared subspace representations is collaboratively explored. And the structural semantics embedded in subspace representations are fed back to latent representation and recovered views, improving the imputation fidelity and clustering discriminability.

- An effective iterative method is designed to solve the optimization problem. Extensive experiments verify the superiority of the proposed ARSL-IMVC method in incomplete clustering task.

## 2 RELATED WORK

Recently, numerous IMVC methods have been widely proposed (Wen et al., 2020a; Shen et al., 2025; Jiang et al., 2025; Li et al., 2024; Wen et al., 2024a; Hu et al., 2025), which could be roughly grouped into two categories according to the way of handling missing samples. The first category crudely ignores missing samples and focuses on learning clustering representation from available views, i.e., imputation-free. Li et al. divided samples into view-complete parts and view-specific missing parts and learned low-dimensional representation by non-negative matrix factorization (Li et al., 2014). Hu et al. proposed doubly aligned incomplete multi-view clustering (DAIMC), aligning the available views to learn a compact representation shared by all views and trying to weaken the influence of missing samples (Hu & Chen, 2018). Due to the powerful relationship representa-

tion ability of graph, the incompleteness of data is transferred to the similarity domain (Wen et al., 2020b; 2023a). Wen et al. designed a graph-based IMVC method, Incomplete Multi-view Spectral Clustering with Adaptive Graph Learning (IMSC-AGC), where partial graph for each view is adaptively constructed by only leveraging observable samples and expanded to complete graph, and then a shared spectral embedding is learned (Wen et al., 2020a). To weaken the impact of noisy view, Wen et al. further designed a highly confident local structure induced consensus graph learning (HCLS-CGL) for IMVC (Wen et al., 2023a). Some methods also tried to maximize the utilization of available information by simultaneously considering feature and graph structure and establishing a connection between them (Bai et al., 2024; Liu et al., 2024b).

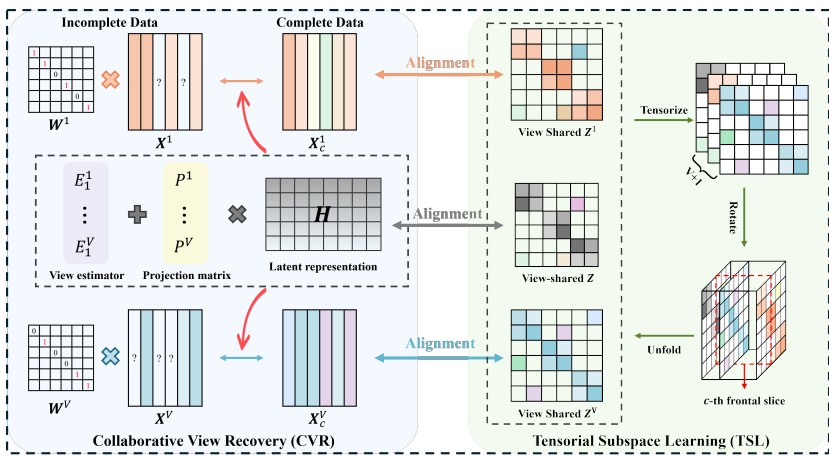

Figure 1: The overall framework of proposed ARSL-IMVC method, which mainly consists of CVR and TSL modules and aligns them in cross-view consistency and complementarity exploration by a latent representation.

Although achieving good progress, the imputation-free methods ignore the underlying connections between samples or views, which instinctively undermines the structural integrity and compromises the clustering performance. To address this issue, imputation-based IMVC methods recover missing data and learn clustering representation on the filling data. Wen et al. proposed a unified embedding alignment framework (UEAF) to infer missing samples, fully leveraging the cross-view correlations by common representation (Wen et al., 2019). Wang et al. leveraged both local view-specific and global view-shared similarity structures to guide the recovery (Wang et al., 2024). To utilize high-order relationships between samples, Chen et al. developed hypergraph induced missing reconstruction strategy for IMVC (IMVC-HG) and further explored the complementarity embedded in inter-view label representations (Chen et al., 2025). Guo et al. proposed the robust mixed-order graph learning (RMoGL), utilizing the mixed-order structural information of recovered data from low-order to high-order (Guo et al., 2025b). In addition, several methods focus on modeling both complex cross-view correlation and similarity relationships between samples by low-rank tensor learning (Li et al., 2022; Yao et al., 2025; Chen et al., 2024; Li et al., 2025). Li et al. explored the high-order correlation of views and that of samples, utilizing hyper-Laplacian regularizer in view recovery and low-rank tensor learning in subspace structure recovery (Li et al., 2022). Yao et al. focused on recovering the multi-view similarity graphs and utilized dual tensor constraint to explore complex correlations, which achieves the between/within view information completing for tensorial incomplete multi-view clustering (BWIC-TIMC) (Yao et al., 2025). In addition, to improve data separability, some kernel-based IMVC methods reconstruct the missing data in Hilbert kernel space (Liu et al., 2020; Wu et al., 2024; Liu et al., 2021; Wu et al., 2025). Although most existing IMVC methods unifying missing view recovery and clustering representation learning have achieved notable progress, they are only treated as weakly coupled strategies. Specifically, they fail to simultaneously exploit the inherent consistency and complementarity among multiple views during the view recovery and representation learning and achieve the semantic correlation exploration alignment through explicit bridge, resulting in shallow collaborative interaction between them and sub-optimal clustering structure. Thus, this study explicitly aligns the collaborative view recovery and tensorial subspace learning by a shared latent representation for IMVC (ARSL-IMVC), bridging the tight interaction between them.

Table 1: Main notations and descriptions in this study.

| Notations | Descriptions | Notations | Descriptions |
|---|---|---|---|
| $\boldsymbol{X}^v \in \mathbb{R}^{d_v \times n}$ | Data matrix of the $v$-th view | $\boldsymbol{E}_1^v$ | View recovery estimator |
| $\boldsymbol{W}^v \in \{0,1\}^{n \times n}$ | Missing indicator matrix | $\boldsymbol{E}_2^v, \boldsymbol{E}_H$ | Noise matrices |
| $\boldsymbol{P}^v \in \mathbb{R}^{d_v \times k}$ | Projection matrix of the $v$-th view | $\boldsymbol{Z}^v, \boldsymbol{Z} \in \mathbb{R}^{n \times n}$ | Specific and shared representations |
| $\boldsymbol{H} \in \mathbb{R}^{k \times n}$ | Latent representation matrix | $\mathcal{Z} \in \mathbb{R}^{n \times (V+1) \times n}$ | Subspace representation tensor |

## 3  METHODOLOGY

In this section, the proposed ARSL-IMVC model is illustrated in detail from two main modules: Collaborative View Recovery (CVR) and Tensorial Subspace Learning (TSL), where the framework is presented in Figure 1. Then, the iterative optimization procedure is provided. The basic notations are conveniently summarized in the Table 1.

### 3.1  COLLABORATIVE VIEW RECOVERY

Consider the incomplete multi-view data consisting of $n$ samples from $V$ views, i.e., $\{\{\boldsymbol{x}_i^v\}_{i=1}^n\}_{v=1}^V$, where $\boldsymbol{x}_i^v \in \mathbb{R}^{d_v}$ is the $i$-th sample in $v$-th view. The incomplete pattern is symbolized with a diagonal indicator matrix $\boldsymbol{W}^v \in \{0,1\}^{n \times n}$ such that $\boldsymbol{W}_{ii}^v = 1$ if $i$-th sample exists in $v$-th view and 0 otherwise. The MVC adheres to a convincing assumption that multi-view data is usually embedded in the shared latent space. With this inverse assumption, the ARSL-IMVC tries to linearly infer missing views from a "virtual" latent representation $\boldsymbol{H} \in \mathbb{R}^{k \times n}$ via reconstruction operator $\boldsymbol{P}^v \in \mathbb{R}^{d_v \times k}$, which is beneficial for reshaping the consistency among multiple views. To improve the freedom of view recovery, view-specific feature estimator $\boldsymbol{E}_1^v$ is introduced and thus the view reconstruction is formulated, i.e.,

$$\boldsymbol{P}^v \boldsymbol{H} + \boldsymbol{E}_1^v \tag{1}$$

Further, to ensure that the various recovered views retain sufficient complementary information, the Hilbert-Schmidt Independence Criterion (HSIC) is introduced as diversity regularizer between any estimator pair $\boldsymbol{E}_1^v$ and $\boldsymbol{E}_1^w$, where the empirical HSIC term is defined (Gretton et al., 2007):

$$\mathrm{HSIC}(\boldsymbol{E}_1^v, \boldsymbol{E}_1^w) = \mathrm{Tr}(\boldsymbol{K}_v \tilde{\boldsymbol{H}} \boldsymbol{K}_w \tilde{\boldsymbol{H}})/(n-1)^2 \tag{2}$$

where $\boldsymbol{K}_v$ and $\boldsymbol{K}_w$ are inner product kernel matrices of $\boldsymbol{E}_1^v$ and $\boldsymbol{E}_1^w$ respectively, $\tilde{\boldsymbol{H}} = \boldsymbol{I}_n - \frac{1}{n}\boldsymbol{1}\boldsymbol{1}^T$ is the centralized matrix. The HSIC term penalizes the dependence between the various reconstruction views, thereby encouraging diversity and informativeness among them in the feature level (Guo et al., 2022). To explore the consistency-diversity and ensure the reconstruction quality, the CVR module is formulated as follows:

$$\min_{\boldsymbol{H}, \boldsymbol{P}^v, \boldsymbol{E}_1^v} \sum_{w=1; w \neq v}^{V} \mathrm{HSIC}(\boldsymbol{E}_1^v, \boldsymbol{E}_1^w)$$
$$\text{s.t. } \boldsymbol{X}^v \boldsymbol{W}^v = (\boldsymbol{P}^v \boldsymbol{H} + \boldsymbol{E}_1^v)\boldsymbol{W}^v, (\boldsymbol{P}^v)^T \boldsymbol{P}^v = \boldsymbol{I} \tag{3}$$

where $\boldsymbol{X}^v = [\boldsymbol{x}_1^v, \boldsymbol{x}_2^v, \cdots, \boldsymbol{x}_n^v]$, $\boldsymbol{X}^v \boldsymbol{W}^v$ denotes the non-missing data, and the equality constraint is enforced to ensure reconstruction fidelity of observable samples.

### 3.2  TENSORIAL SUBSPACE LEARNING

Subspace representation learning is an effective method for exploring clustering semantics in low-dimensional embedding space, especially for self-representation learning methods. Leveraging the shared latent representation and recovered view-specific features, the global semantics and local clustering structures could be effectively characterized, i.e.,

$$\boldsymbol{H} = \boldsymbol{H}\boldsymbol{Z} + \boldsymbol{E}_H, \boldsymbol{P}^v \boldsymbol{H} + \boldsymbol{E}_1^v = (\boldsymbol{P}^v \boldsymbol{H} + \boldsymbol{E}_1^v)\boldsymbol{Z}^v + \boldsymbol{E}_2^v \tag{4}$$

where $\boldsymbol{Z}, \boldsymbol{Z}^v$ are view-shared and view-specific subspace representations, respectively, encoding the comprehensive structural semantics; and $\boldsymbol{E}_H$ and $\boldsymbol{E}_2^v$ are noise terms.

To explore the consistency and complementarity across views in subspace representation level, both shared and specific subspace representations into a unified low-rank tensor. Thus, the TSL module is formulated as follows:

$$\min_{\boldsymbol{Z},\boldsymbol{Z}^v,\boldsymbol{E}_H,\boldsymbol{E}_2^v} \|\mathcal{Z}\|_{\circledast} + \lambda_1(\|\boldsymbol{E}_H\|_{2,1} + \sum_{v=1}^{V} \|\boldsymbol{E}_2^v\|_{2,1})$$
$$\text{s.t. } \boldsymbol{H} = \boldsymbol{HZ} + \boldsymbol{E}_H, \boldsymbol{P}^v\boldsymbol{H} + \boldsymbol{E}_1^v = (\boldsymbol{P}^v\boldsymbol{H} + \boldsymbol{E}_1^v)\boldsymbol{Z}^v + \boldsymbol{E}_2^v, \quad (5)$$
$$\mathcal{Z} = \Phi(\boldsymbol{Z}^1, \boldsymbol{Z}^2, \cdots, \boldsymbol{Z}^V, \boldsymbol{Z})$$

where $\Phi(\cdot)$ is a tensor construction function, which stacks the representations $(\{\boldsymbol{Z}^v\}_{v=1}^V, \boldsymbol{Z})$ and rotates to form a subspace representation tensor $\mathcal{Z} \in \mathbb{R}^{n \times (V+1) \times n}$. In the low-rank tensor space, the high-order cross-view correlations at different levels be effectively captured, facilitating semantic alignment across diverse views and achieving collaborative interaction between local and global structural information.

## 3.3 FORMULATION OF THE PROPOSED ARSL-IMVC

By incorporating the CVR and TSL into a unified framework, the final objective of ARSL-IMVC is formulated as:

$$\min_{\Upsilon} \|\mathcal{Z}\|_{\circledast} + \lambda_1(\|\boldsymbol{E}_H\|_{2,1} + \sum_{v=1}^{V} \|\boldsymbol{E}_2^v\|_{2,1}) + \lambda_2 \sum_{w=1;w\neq v}^{V} \text{HSIC}(\boldsymbol{E}_1^v, \boldsymbol{E}_1^w)$$
$$\text{s.t. } \boldsymbol{X}^v\boldsymbol{W}^v = (\boldsymbol{P}^v\boldsymbol{H} + \boldsymbol{E}_1^v)\boldsymbol{W}^v, \boldsymbol{P}^v\boldsymbol{H} + \boldsymbol{E}_1^v = (\boldsymbol{P}^v\boldsymbol{H} + \boldsymbol{E}_1^v)\boldsymbol{Z}^v + \boldsymbol{E}_2^v, \quad (6)$$
$$\boldsymbol{H} = \boldsymbol{HZ} + \boldsymbol{E}_H, (\boldsymbol{P}^v)^T\boldsymbol{P}^v = \boldsymbol{I}, \mathcal{Z} = \Phi(\boldsymbol{Z}^1, \boldsymbol{Z}^2, \cdots, \boldsymbol{Z}^V, \boldsymbol{Z})$$

where $\Upsilon = \{\boldsymbol{H}, \boldsymbol{P}^v, \boldsymbol{E}_1^v, \boldsymbol{Z}, \boldsymbol{E}_H, \boldsymbol{Z}^v, \boldsymbol{E}_2^v\}$ is unknown variable set, $\lambda_1$ and $\lambda_2$ are hyper-parameters that control the contribution of different regularization terms in the objective function. In Eq.(6), the shared latent representation $\boldsymbol{H}$ not only facilitates the joint optimization and collaborative interaction between view recovery and subspace learning but also serves as a semantic anchor that aligns the reconstructed views with corresponding subspace representations in capturing cross-view complementarity and consistency. The information flow propagation enables coherent cross-view semantic correlation exploration, promoting the clustering quality for complex IMVC task.

## 3.4 OPTIMIZATION

The objective function in Eq.(6) with multiple variables is difficult to solve directly, the Alternating Direction Method of Multipliers (ADMM) (Lin et al., 2011) is utilized to iteratively optimize each variable. To make the objective function separable, auxiliary variables $\mathcal{J}$ and $\boldsymbol{X}_c^v$ are introduced, and then the augmented Lagrange function is defined:

$$\mathcal{L}(\Upsilon, \boldsymbol{X}_c^v, \mathcal{J}; \boldsymbol{Y}_1^v, \boldsymbol{Y}_2^v, \boldsymbol{Y}_3^v, \boldsymbol{Y}_4, \mathcal{Y}, \mu) = \|\mathcal{J}\|_{\circledast} + \lambda_1(\|\boldsymbol{E}_H\|_{2,1} + \sum_{v=1}^{V} \|\boldsymbol{E}_2^v\|_{2,1})$$

$$+ \lambda_2 \sum_{w=1;v\neq w}^{V} \text{HSIC}(\boldsymbol{E}_1^v, \boldsymbol{E}_1^w) + \sum_{v=1}^{V} \phi(\boldsymbol{Y}_1^v, \boldsymbol{X}_c^v - \boldsymbol{P}^v\boldsymbol{H} - \boldsymbol{E}_1^v)$$

$$+ \sum_{v=1}^{V} \phi(\boldsymbol{Y}_2^v, \boldsymbol{X}^v\boldsymbol{W}^v - \boldsymbol{X}_c^v\boldsymbol{W}^v) + \sum_{v=1}^{V} \phi(\boldsymbol{Y}_3^v, \boldsymbol{X}_c^v - \boldsymbol{X}_c^v\boldsymbol{Z}^v - \boldsymbol{E}_2^v)$$

$$+ \phi(\boldsymbol{Y}_4, \boldsymbol{H} - \boldsymbol{HZ} - \boldsymbol{E}_H) + \phi(\mathcal{Y}, \mathcal{Z} - \mathcal{J}) \quad (7)$$

where $\{\boldsymbol{Y}_1^v, \boldsymbol{Y}_2^v, \boldsymbol{Y}_3^v, \boldsymbol{Y}_4, \mathcal{Y}\}$ are Lagrangian multipliers, $\phi(\boldsymbol{A}, \boldsymbol{B}) = \frac{\mu}{2}\|\boldsymbol{B}\|_F^2 + \langle \boldsymbol{A}, \boldsymbol{B} \rangle$, $\mu$ is a penalty factor. Then, each variable is alternately updated as follows:

**Update $\boldsymbol{P}^v$**: Fixing other variables except $\boldsymbol{P}^v$, the subproblem in Eq.(7) w.r.t. $\boldsymbol{P}^v$ is as follows:

$$\min_{(\boldsymbol{P}^v)^T\boldsymbol{P}^v=\boldsymbol{I}} \phi(\boldsymbol{Y}_1^v, \boldsymbol{X}_c^v - \boldsymbol{P}^v\boldsymbol{H} - \boldsymbol{E}_1^v) \iff \min_{(\boldsymbol{P}^v)^T\boldsymbol{P}^v=\boldsymbol{I}} Tr((\boldsymbol{P}^v)^T(\boldsymbol{X}_c^v - \boldsymbol{E}_1^v + \boldsymbol{Y}_1^v/\mu)\boldsymbol{H}^T)$$
$$(8)$$

The optimal closed-form solution could be obtained (Wang et al., 2019), i.e., $\boldsymbol{P}^v = \boldsymbol{U}\boldsymbol{V}^T$, where $\boldsymbol{U}$ and $\boldsymbol{V}$ are the left and right singular vector of $(\boldsymbol{X}_c^v - \boldsymbol{E}_1^v + \boldsymbol{Y}_1^v/\mu)\boldsymbol{H}^T$.

**Update $\boldsymbol{X}_c^v$**: Fixing other variables except $\boldsymbol{X}_c^v$, the subproblem in Eq.(7) w.r.t. $\boldsymbol{X}_c^v$ is as follows:

$$\min_{\boldsymbol{X}_c^v} \phi(\boldsymbol{Y}_1^v, \boldsymbol{X}_c^v - \boldsymbol{P}^v\boldsymbol{H} - \boldsymbol{E}_1^v) + \phi(\boldsymbol{Y}_2^v, \boldsymbol{X}^v\boldsymbol{W}^v - \boldsymbol{X}_c^v\boldsymbol{W}^v) + \phi(\boldsymbol{Y}_3^v, \boldsymbol{X}_c^v - \boldsymbol{X}_c^v\boldsymbol{Z}^v - \boldsymbol{E}_2^v) \quad (9)$$

Taking the derivative w.r.t. $\boldsymbol{X}_c^v$ and setting it to zero, the optimal solution for $\boldsymbol{X}_c^v$ is as follows:

$$\boldsymbol{X}_c^v = \big(\boldsymbol{A}_1 + \boldsymbol{A}_2(\boldsymbol{W}^v)^T + \boldsymbol{A}_3(\boldsymbol{I} - \boldsymbol{Z}^v)^T\big)\big(\boldsymbol{I} + \boldsymbol{W}^v(\boldsymbol{W}^v)^T + (\boldsymbol{I} - \boldsymbol{Z}^v)(\boldsymbol{I} - \boldsymbol{Z}^v)^T\big)^{-1} \quad (10)$$

where $\boldsymbol{A}_1 = \boldsymbol{P}^v\boldsymbol{H} + \boldsymbol{E}_1^v - \boldsymbol{Y}_1^v/\mu$, $\boldsymbol{A}_2 = \boldsymbol{X}^v\boldsymbol{W}^v + \boldsymbol{Y}_2^v/\mu$, and $\boldsymbol{A}_3 = \boldsymbol{E}_2^v - \boldsymbol{Y}_3^v/\mu$.

**Update $\boldsymbol{H}$**: Fixing other variables except $\boldsymbol{H}$, the subproblem in Eq.(7) w.r.t. $\boldsymbol{H}$ is as follows:

$$\min_{\boldsymbol{H}} \sum_{v=1}^{V} \phi(\boldsymbol{Y}_1^v, \boldsymbol{X}_c^v - \boldsymbol{P}^v\boldsymbol{H} - \boldsymbol{E}_1^v) + \phi(\boldsymbol{Y}_4, \boldsymbol{H} - \boldsymbol{H}\boldsymbol{Z} - \boldsymbol{E}_H) \quad (11)$$

Taking the derivative w.r.t. variable $\boldsymbol{H}$ and setting it to zero, the optimal solution could be obtained:

$$\boldsymbol{A}\boldsymbol{H} + \mu\boldsymbol{H}\boldsymbol{B} = \boldsymbol{C} \quad (12)$$

where $\boldsymbol{A} = \sum_{v=1}^{V} \mu(\boldsymbol{P}^v)^T\boldsymbol{P}^v$, $\boldsymbol{B} = \mu(\boldsymbol{I} - \boldsymbol{Z} - \boldsymbol{Z}^T + \boldsymbol{Z}\boldsymbol{Z}^T)$, $\boldsymbol{C} = \mu\left(\boldsymbol{E}_H - \boldsymbol{Y}_4/\mu\right)\left(\boldsymbol{I} - \boldsymbol{Z}^T\right) + \sum_{v=1}^{V} \mu(\boldsymbol{P}^v)^T\left(\boldsymbol{X}_c^v - \boldsymbol{E}_1^v + \boldsymbol{Y}_1^v/\mu\right)$. It is a Sylvester equation in Eq.(12) (Bartels & Stewart, 1972). The matrix $\boldsymbol{A}$ is relaxed into $\hat{\boldsymbol{A}} = \boldsymbol{A} + \epsilon\boldsymbol{I}$ with strictly positive definiteness for the solution stability ($\epsilon$ is a small positive scalar).

**Update $\boldsymbol{Z}^v$**: Fixing other variables except $\boldsymbol{Z}^v$, the subproblem in Eq.(7) w.r.t $\boldsymbol{Z}^v$ is as follows:

$$\min_{\boldsymbol{Z}^v} \phi(\boldsymbol{Y}_3^v, \boldsymbol{X}_c^v - \boldsymbol{X}_c^v\boldsymbol{Z}^v - \boldsymbol{E}_2^v) + \phi(\boldsymbol{Y}^v, \boldsymbol{Z}^v - \boldsymbol{J}^v) \quad (13)$$

where $\boldsymbol{Y}^v = \Phi_v^{-1}(\mathcal{Y})$, $\boldsymbol{Z}^v = \Phi_v^{-1}(\mathcal{Z})$ and $\boldsymbol{J}^v = \Phi_v^{-1}(\mathcal{J})$. Taking the derivative w.r.t. $\boldsymbol{Z}^v$ and setting it to zero, the optimal solution for variable $\boldsymbol{Z}^v$ could be obtained:

$$\boldsymbol{Z}^v = ((\boldsymbol{X}_c^v)^T\boldsymbol{X}_c^v + \boldsymbol{I})^{-1}\big((\boldsymbol{X}_c^v)^T\boldsymbol{X}_c^{(v)} - (\boldsymbol{X}_c^v)^T\boldsymbol{E}_2^v + (\boldsymbol{X}_c^v)^T\boldsymbol{Y}_3^v/\mu + \boldsymbol{J}^v - \boldsymbol{Y}^v/\mu\big) \quad (14)$$

**Update $\boldsymbol{Z}$**: Fixing other variables except $\boldsymbol{Z}$, the subproblem in Eq.(7) w.r.t. $\boldsymbol{Z}$ is as follows:

$$\min_{\boldsymbol{Z}} \phi(\boldsymbol{Y}_4, \boldsymbol{H} - \boldsymbol{H}\boldsymbol{Z} - \boldsymbol{E}_H) + \phi(\boldsymbol{Y}, \boldsymbol{Z} - \boldsymbol{J}) \quad (15)$$

where $\boldsymbol{Y} = \Phi_{V+1}^{-1}(\mathcal{Y})$, $\boldsymbol{Z} = \Phi_{V+1}^{-1}(\mathcal{Z})$ and $\boldsymbol{J} = \Phi_{V+1}^{-1}(\mathcal{J})$. Taking the derivative w.r.t. $\boldsymbol{Z}$ and setting it to zero, the optimal solution for variable $\boldsymbol{Z}$ could be obtained:

$$\boldsymbol{Z} = (\boldsymbol{H}^T\boldsymbol{H} + \boldsymbol{I})^{-1}\big(\boldsymbol{H}^T(\boldsymbol{H} - \boldsymbol{E}_H + \boldsymbol{Y}_4/\mu) + \boldsymbol{J} - \boldsymbol{Y}/\mu\big) \quad (16)$$

**Update $\boldsymbol{E}_1^v$**: Fixing other variables except $\boldsymbol{E}_1^v$, the subproblem in Eq.(7) w.r.t. $\boldsymbol{E}_1^v$ is as follows:

$$\min_{\boldsymbol{E}_1^v} \phi(\boldsymbol{Y}_1^v, \boldsymbol{X}_c^v - \boldsymbol{P}^v\boldsymbol{H} - \boldsymbol{E}_1^v) + \lambda_2 \sum_{w=1;w\neq v}^{V} \text{HSIC}(\boldsymbol{E}_1^v, \boldsymbol{E}_1^w) \quad (17)$$

Here, the inner product kernel (i.e., $\boldsymbol{K} = (\boldsymbol{E}_1^v)^T\boldsymbol{E}_1^v$) is utilized for HSIC term. Taking the derivative w.r.t. $\boldsymbol{E}_1^v$ and setting it to zero, the optimal solution for variable $\boldsymbol{E}_1^v$ could be obtained:

$$\boldsymbol{E}_1^v = \big(\boldsymbol{X}_c^v - \boldsymbol{P}^v\boldsymbol{H} + \boldsymbol{Y}_1^v/\mu\big)\Big(\boldsymbol{I} + \frac{2\lambda_2}{\mu(n-1)^2} \sum_{w=1;w\neq v}^{V} \tilde{\boldsymbol{H}}\boldsymbol{K}^w\tilde{\boldsymbol{H}}\Big)^{-1} \quad (18)$$

**Update $\boldsymbol{E}_2^v$**: Fixing other variables except $\boldsymbol{E}_2^v$, the subproblem in Eq.(7) w.r.t. $\boldsymbol{E}_2^v$ is as follows:

$$\min_{\boldsymbol{E}_2^v} \lambda_1\|\boldsymbol{E}_2^v\|_{2,1} + \phi(\boldsymbol{Y}_3^v, \boldsymbol{X}_c^v - \boldsymbol{X}_c^v\boldsymbol{Z}^v - \boldsymbol{E}_2^v) \iff \min_{\boldsymbol{E}_2^v} \lambda_1/\mu\|\boldsymbol{E}_2^v\|_{2,1} + 1/2\|\boldsymbol{E}_2^v - \boldsymbol{L}^v\|_F^2 \quad (19)$$

where $\boldsymbol{L}^v = \boldsymbol{X}_c^v - \boldsymbol{X}_c^v\boldsymbol{Z}^v + \boldsymbol{Y}_3^v/\mu$. Its solution can be obtained by $l_{2,1}$ minimization thresholding operator column by column (Liu et al., 2010), i.e.,

$$\boldsymbol{E}_2^v(:,j) = \Big(1 - \frac{\lambda_1}{\mu\,\|\boldsymbol{L}^v(:,j)\|_2}\Big)^{+} \boldsymbol{L}^v(:,j) \quad (20)$$

where $(x)^+ = \max(x, 0)$.

**Update $E_H$:** Fixing other variables except $E_H$, the subproblem in Eq.(7) w.r.t. $E_H$ is as follows:

$$\min_{E_H} \lambda_1 \|E_H\|_{2,1} + \phi(Y_4, H - HZ + Y_4/\mu) \tag{21}$$

Similar to $E_2^v$, the solution of $E_H$ could be solved by $l_{2,1}$ minimization thresholding operator column by column, i.e.,

$$E_H(:,j) = \left(1 - \frac{\lambda_1}{\mu \|M(:,j))\|_2}\right)^+ M(:,j) \tag{22}$$

where $M = H - HZ + Y_4/\mu$.

**Update $\mathcal{J}$:** Fixing other variables except $\mathcal{J}$, the subproblem in Eq.(7) w.r.t. variable $\mathcal{J}$ is as follows:

$$\min_{\mathcal{J}} \|\mathcal{J}\|_\circledast + \phi(\mathcal{Y}, \mathcal{Z} - \mathcal{J}) \iff \min_{\mathcal{J}} 1/\mu \|\mathcal{J}\|_\circledast + 1/2\|\mathcal{Z} + \mathcal{Y}/\mu - \mathcal{J}\|_F^2 \tag{23}$$

It is a classical tensor nuclear norm minimization problem, where the closed-form solution could be obtained via tensor singular value thresholding (t-SVT) (Zhang et al., 2014).

**Update multipliers and penalty parameter:** The $\{Y_1^v, Y_2^v, Y_3^v\}_{v=1}^V$, $Y_4$, $\mathcal{Y}$, $\mu$ are updated by:

$$\begin{cases} Y_1^v := Y_1^v + \mu(X_c^v - P^v H - E_1^v), Y_2^v := Y_2^v + \mu(X^v W^v - X_c^v W^v), \\ Y_3^v := Y_3^v + \mu(X_c^v - X_c^v Z^v - E_2^v), Y_4 := Y_4 + \mu(H - HZ - E_H), \\ \mathcal{Y} := \mathcal{Y} + \mu(\mathcal{Z} - \mathcal{J}), \mu := \min(\rho\mu, \mu_{max}) \end{cases} \tag{24}$$

where $\rho > 1$ is set to accelerate the convergence. With all initial variables, each variable is iteratively updated until stop conditions are satisfied. After obtaining comprehensive subspace representations $(Z, \{Z^v\}_{v=1}^V)$, a powerful affinity $S$ is constructed for spectral clustering, i.e., $S = (|Z| + |Z^T| + \sum_{v=1}^V |Z^v| + \sum_{v=1}^V |(Z^v)^T|)/(V+1)$.

# 4 EXPERIMENTS

## 4.1 EXPERIMENTAL SETTING

***Datasets:*** To verify the effectiveness of ARSL-IMVC method, seven benchmark datasets are utilized, including BBCSport (Zhang et al., 2024), HW (Asuncion et al., 2007), BDGP (Cai et al., 2012), Yale (Guo et al., 2025b), NGs (Hussain et al., 2010), 100leaves (Mallah et al., 2013), and Scene-15 (Li & Perona, 2005). ***Competitors:*** Several representative methods are selected as competitors, including **BSV** (Ng et al., 2001), **Concat** (Wen et al., 2023b), **IMSC-AGL** (Wen et al., 2020a), **DAIMC** (Hu & Chen, 2018), **UEAF** (Wen et al., 2019), **HCP-IMSC** (Li et al., 2022), **HCLS-CGL** (Wen et al., 2023a), **BWIC-TIMC** (Yao et al., 2025), **RMoGL** (Guo et al., 2025b). ***Incomplete Data Construction:*** The incomplete multi-view data is constructed by randomly removing samples from each view, with missing rates $p \in \{0.1, 0.3, 0.5\}$ on the BBCSport, HW, BDGP datasets, missing rates $p \in \{0.1 : 0.1 : 0.8\}$ on the Yale, NGs, 100leaves, and Scene-15 datasets. Following the experimental setting in (Liu et al., 2024b), each sample appears in at least one view. ***Parameters Settings:*** All compared methods are implemented with their public source codes and recommended parameter settings. For ARSL-IMVC, both $\lambda_1$ and $\lambda_2$ are searched within the range of $\{1, 10, 20, 30, 40, 50\}$, and $k$ is selected between 10 and 20. ***Clustering Metrics:*** To make comprehensive comparison, all methods are repeated ten times and the average values of Accuracy (ACC), Normalized Mutual Information (NMI), and Purity metrics are reported.

## 4.2 EXPERIMENTAL RESULTS AND ANALYSIS

Table 2 presents quantitative clustering results on the BBCSport, HW, and BDGP datasets, where the best and sub-optimal results are marked in bold and underline. Figure 2 illustrates the experimental results with broader missing rates on the Yale, NGs, 100leaves and Scene-15 datasets. The following conclusions can be concluded:

- The proposed ARSL-IMVC method consistently outperforms other IMVC existing methods on most cases. And, the ARSL-IMVC achieves significant performance improvements. For exapmle, in terms of ACC, it improves around $4.60\%$, $8.31\%$, and $5.41\%$ over sub-optimal methods respectively on BBCSport, HW, and BDGP datasets, when missing ratio is 0.1.

Table 2: Clustering results of all methods on the BBCSport, HW, and BDGP datasets.

| Dataset | $p$ | Metrics | BSV | Concat | DAIMC | UEAF | IMSC-AGL | HCLS-CGL | HCP-IMSC | BWIC-TIMC | RMoGL | **Ours** |
|---|---|---|---|---|---|---|---|---|---|---|---|---|
| BBCSport | 0.1 | ACC | 35.85 | 58.07 | 75.39 | 71.32 | 84.30 | 72.00 | _91.91_ | 90.75 | 89.19 | **96.51** |
| | | NMI | 1.98 | 34.96 | 60.76 | 63.27 | 72.63 | 68.90 | 79.84 | 82.10 | _83.18_ | **89.77** |
| | | Purity | 36.21 | 59.52 | 78.18 | 80.70 | 85.75 | 75.55 | _91.91_ | 90.34 | 88.79 | **96.51** |
| | 0.3 | ACC | 33.82 | 46.27 | 75.31 | 77.39 | 88.27 | 79.96 | _89.15_ | 78.94 | 78.49 | **94.85** |
| | | NMI | 1.82 | 17.69 | 58.29 | 58.19 | 74.34 | 71.33 | _75.47_ | 70.12 | 68.86 | **84.95** |
| | | Purity | 36.21 | 47.81 | 77.46 | 79.78 | 88.62 | 83.45 | _89.15_ | 81.25 | 81.80 | **94.85** |
| | 0.5 | ACC | 30.51 | 45.44 | 57.74 | 67.22 | 81.99 | 78.49 | _86.05_ | 77.53 | 76.47 | **88.97** |
| | | NMI | 2.32 | 15.81 | 41.25 | 52.05 | 66.79 | 67.13 | **72.37** | 65.84 | 61.76 | _71.32_ |
| | | Purity | 36.21 | 46.69 | 66.62 | 73.29 | 82.52 | 82.17 | _86.05_ | 79.48 | 79.60 | **88.97** |
| HW | 0.1 | ACC | 44.10 | 65.08 | 76.65 | 53.79 | _88.59_ | 79.81 | 79.80 | 81.85 | 76.73 | **96.90** |
| | | NMI | 52.11 | 63.83 | 71.59 | 50.04 | _87.21_ | 81.79 | 75.73 | 82.87 | 74.61 | **92.77** |
| | | Purity | 44.20 | 68.99 | 78.99 | 54.01 | _90.67_ | 81.97 | 80.05 | 81.85 | 76.73 | **96.90** |
| | 0.3 | ACC | 38.50 | 57.05 | 59.18 | 44.48 | _84.28_ | 81.45 | 75.35 | 74.75 | 64.19 | **92.46** |
| | | NMI | 43.11 | 52.25 | 51.45 | 40.66 | 78.31 | _81.04_ | 69.11 | 72.63 | 62.25 | **84.16** |
| | | Purity | 38.90 | 58.97 | 60.16 | 45.35 | _85.64_ | 84.00 | 76.50 | 74.75 | 65.03 | **92.46** |
| | 0.5 | ACC | 31.75 | 44.86 | 62.59 | 37.07 | 79.43 | _81.75_ | 70.80 | 60.13 | 59.20 | **89.03** |
| | | NMI | 33.78 | 39.15 | 51.01 | 30.85 | 73.36 | **81.39** | 60.47 | 59.70 | 54.74 | _77.97_ |
| | | Purity | 32.00 | 47.15 | 63.28 | 37.27 | 81.34 | _84.00_ | 71.30 | 62.64 | 59.35 | **89.03** |
| BDGP | 0.1 | ACC | 40.28 | 45.02 | 45.72 | _50.66_ | 41.67 | 23.68 | 21.08 | 29.88 | 45.94 | **56.07** |
| | | NMI | 25.22 | 22.13 | 22.91 | _28.03_ | 19.90 | 3.37 | 25.26 | 7.23 | 23.48 | **35.22** |
| | | Purity | 46.08 | 46.58 | 47.75 | _52.42_ | 44.57 | 23.68 | 19.52 | 31.04 | 46.66 | **56.07** |
| | 0.3 | ACC | 39.22 | 39.20 | 41.85 | _46.82_ | 38.97 | 23.84 | 23.93 | 20.32 | 42.66 | **50.58** |
| | | NMI | 23.31 | 16.55 | 17.03 | 22.58 | 16.93 | 3.36 | _29.06_ | 1.30 | 20.30 | **31.59** |
| | | Purity | 43.52 | 40.71 | 42.98 | _49.86_ | 41.55 | 23.84 | 22.21 | 20.36 | 43.60 | **52.07** |
| | 0.5 | ACC | 37.68 | 34.90 | 39.74 | _45.92_ | 37.59 | 24.28 | 20.46 | 25.52 | 31.68 | **49.21** |
| | | NMI | 20.99 | 11.78 | 20.22 | 24.14 | 15.81 | 3.57 | _24.84_ | 2.43 | 6.77 | **32.16** |
| | | Purity | 40.60 | 35.43 | 41.87 | _47.78_ | 39.42 | 24.32 | 19.00 | 25.61 | 32.68 | **50.21** |

Figure 2: Clustering results on Yale, NGs, 100leaves and Scene-15 with different missing rates.

- Compared to imputation-free IMVC methods (DAIMC, IMSC-AGL, HCLS-CGL), the proposed ARSL-IMVC usually achieves superior clustering performance, illustrating the effectiveness of proposed collaborative view recovery strategy. And, it highlights that the reshaping cross-view consistency and diversity in feature-level indeed facilitates credible and strong view recovery.

- The proposed ARSL-IMVC method also outperforms the imputation-based methods (i.e., UEAF, HCP-IMSC, BWIC-TIMC, and RMoGL), owing to its learned latent representation that simultaneously serves as a semantic foundation for both view recovery and subspace learning, thereby enabling their alignment in capturing cross-view consistency and complementarity. This explicit information flow transmission promotes deep interaction between CVR and TSL modules, improving the ability to accurately recover missing view and clustering semantic discriminability.

- As illustrated in Figure 2, most IMVC methods exhibit significant performance degradation as the missing rate increases, while ARSL-IMVC maintains higher stability. It fully demonstrates the superiority of proposed ARSL-IMVC in complex IMVC tasks.

## 4.3 ABLATION STUDY

To verify effectiveness of aligning the view recovery and subspace learning by latent representation $H$, one ablation variant ARSL-IMVC-1 is designed by removing the subspace learning on $H$.

As shown in the Table 3, ARSL-IMVC consistently outperforms ablation variant ARSL-IMVC-1, achieving a performance improvement of 12.48%, 26.75%, 9.51%, 6.24%, and 10.63% on BBC-

Table 3: The experimental results of ARSL-IMVC and its ablation variant with the 0.1 missing ratio.

| Datasets | BBCSport | | HW | | Yale | | NGs | | 100leaves | |
|---|---|---|---|---|---|---|---|---|---|---|
| Metrics | ACC | NMI | ACC | NMI | ACC | NMI | ACC | NMI | ACC | NMI |
| ARSL-IMVC-1 | 84.03 | 71.52 | 70.15 | 61.77 | 76.55 | 78.88 | 89.96 | 76.65 | 78.61 | 91.09 |
| ARSL-IMVC | **96.51** | **89.77** | **96.90** | **92.77** | **86.06** | **87.96** | **96.20** | **89.27** | **89.24** | **96.65** |

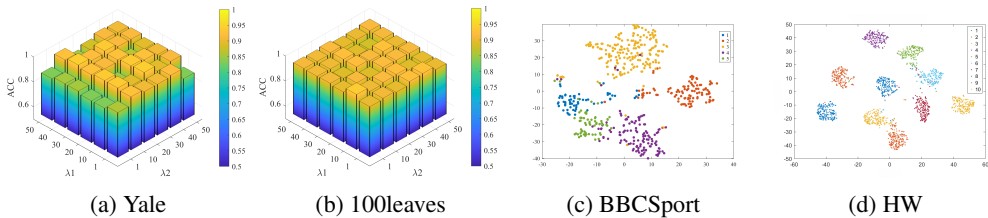

| (a) Yale | (b) 100leaves | (c) BBCSport | (d) HW |
|---|---|---|---|

Figure 3: Parameter sensitivity analysis of the $\lambda_1$ and $\lambda_2$ on Yale and 100leaves datasets with the missing rate of 0.1 in Figures (3a) and (3b). The t-SNE visualization of clustering representation on BBCSport and HW with the missing rate of 0.1 in Figures (3c) and (3d).

Sport, HW, Yale, NGs, and 100leaves datasets regarding ACC metric. The result indicates that aligning view recovery and subspace learning in complex correlation exploration promotes the semantic coherence between them, facilitating the fine-grained and clear clustering structure modeling.

## 4.4 PARAMETER SENSITIVITY AND VISUALIZATION ANALYSIS

To evaluate the sensitivity of ARSL-IMVC to parameter, with a missing rate of 0.1, the ACC metric with different $\lambda_1$ and $\lambda_2$ is shown in Figures (3a) and (3b). It can be observed that the performance of ARSL-IMVC is not significantly influenced by $\lambda_2$ when $\lambda_1$ is fixed, while it is slightly affected by $\lambda_1$ when $\lambda_2$ is fixed. Although, the ARSL-IMVC is relatively robust to parameter varying within a reasonable range.

To intuitively illustrate the clustering performance of ARSL-IMVC, the spectral embeddings on BBCSport and HW datasets are visualized by t-SNE (Maaten & Hinton, 2008) with a missing rate of 0.1. As shown in Figures (3c) and (3d), the spectral embedding obtained by ARSL-IMVC exhibits relatively clear clustering structure and samples from diverse clusters are obviously separated, verifying its discriminability in clustering semantics exploration.

Table 4: Clustering results of some methods on **HDigit** datasets with the missing rate of 0.1.

| Methods | DAIMC | UEAF | IMSC-AGL | HCLS-IMSC | HCP-IMSC | **Ours** |
|---|---|---|---|---|---|---|
| ACC | 67.58 | 85.38 | 76.32 | 98.29 | 89.56 | **99.00** |
| NMI | 64.25 | 73.56 | 77.40 | 95.30 | 89.52 | **96.97** |
| Purity | 69.61 | 85.38 | 77.89 | 98.29 | 88.40 | **99.00** |

## 4.5 SCALABILITY ON LARGE-SCALE DATASET

To further validate the effectiveness and scalability of the proposed ARSL-IMVC on large-scale multi-view data, the Handwritten Digits (HDigit) dataset with 10000 samples is utilized, where two views are collected from various resources: MNIST and USPS. The experimental results of partial representative methods in 0.1 missing ratio are shown in Table 4. Compared with the suboptimal HCLS-IMSC, improvements of approximately 0.7% in both ACC and Purity and approximately 1.7% in NMI are achieved by the proposed method. Compared to other baselines such as UEAF and IMSC-AGL, the performance gains reach 10% ∼ 25%. These results clearly demonstrate that ARSL-IMVC can effectively handle the representation learning challenges arising from substantial increases in sample size and feature dimensionality.

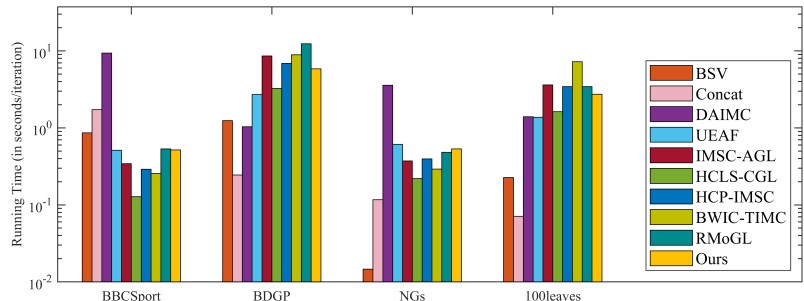

Figure 4: The running time comparisons on the partial representative datasets with 0.1 missing rate.

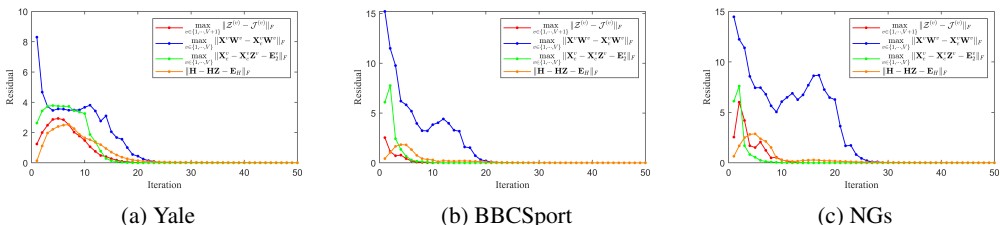

| (a) Yale | (b) BBCSport | (c) NGs |

Figure 5: The convergence curves on Yale, BBCSport and NGs datsets with the missing rate of 0.1.

## 4.6 RUNNING TIME COMPARISON AND CONVERGENCE ANALYSIS

Figure 4 reports the running time of all compared methods on the four benchmark datasets (BBC-Sport, BDGP, NGs and 100leaves). As shown in the Figure 4, the proposed ARSL-IMVC method generally requires comparable computational costs to other IMVC methods on the four datasets. Considering the superior clustering performance of proposed method, ARSL-IMVC demonstrates good balance between computational efficiency and clustering performance.

To empirically shown the stability of the optimization algorithm, the convergence analysis results are verified on the Yale, BBCSport, and NGs datasets with the missing rate of 0.1, where the iterative residual errors are visualized. As shown in Figures (5a), (5b) and (5c), the residual curves of variable updating show that proposed optimization algorithm can reach a local minimum within a finite number of iterations, verifying the fast convergence and numerical stability of ARSL-IMVC.

## 5 CONCLUSION

In this study, a novel ARSL-IMVC method was proposed for challenging incomplete multi-view clustering, which centers on a latent representation to achieve unified alignment between view recovery and tensor subspace learning in complex cross-view correlation exploration. The latent representation served as both view reconstruction basis and global semantics carrier, maintaining and aligning the cross-view consistency. And, global view-shared and local view-specific subspace representations were organized into a low-rank tensor, exploring the cross-view complementarity and multi-level structural correlations. Experimental results demonstrated that the ARSL-IMVC consistently achieves superior clustering performance under various missing rates.

## ACKNOWLEDGMENT

This research was supported by the National Natural Science Foundation of China under Grant 62225303, 62403043, and 62433004; in part by the China Postdoctoral Science Foundation under Grant 2025T180467; in part by the Interdisciplinary Research Center of Beijing University of Chemical Technology under Grant XK2025-06.

ETHICS STATEMENT

Our research adheres strictly to ICLR ethical guidelines, emphasizing responsible AI development to maximize societal benefits while mitigating potential harms. The proposed ARSL-IMVC method enhances incomplete multi-view clustering, offering advancements in data analysis that could benefit fields such as healthcare, environmental monitoring, and education by improving resilience to missing data. Our research is dedicated to the benefit of society and human well-being, and since it does not involve human subjects, no ethical approval is required. All datasets used in our experiments are publicly available and comply with their respective licenses. The methodology and results are presented with full transparency to support reproducibility, including source code provided in the supplementary materials, which will be open-sourced following the review process.

REPRODUCIBILITY STATEMENT

To ensure the reproducibility of our work, the complete source code of the ARSL-IMVC will be provided in the supplementary materials. The public datasets used in the experiments are all cited in the paper and the construction of incomplete datasets can be carried out according to the specific methods mentioned in Section 4.1 of the main text.

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

## A   APPENDIX

**The Use of Large Language Models (LLMs):**  In this research, large language models (LLMs) were employed exclusively to improve the clarity and grammatical accuracy of the manuscript. Their use was limited to refining sentence structure and correcting syntax to enhance readability and professionalism. At no stage did these tools influence the scientific content, methodology, or results. All core ideas and analyses presented are the original work of the authors, with no LLM-generated content contributing to the intellectual substance of the paper.

