# OpenReview forum: "Aligning Collaborative View Recovery and Tensorial Subspace Learning via Latent Representation for Incomplete Multi-View Clustering"
_ICLR.cc/2026/Conference — ICLR 2026 Poster_

### Official Review · Reviewer_WN4G · 2025-10-23

**Soundness:** 3
**Presentation:** 4
**Contribution:** 3
**Rating:** 6
**Confidence:** 4

**Summary:**

This paper proposes a novel method, ARSL-IMVC, to tackle the task of incomplete multi-view clustering. This method combines collaborative view recovery (CVR) and tensor subspace learning (TSL) through a shared latent representation, aiming to enhance cross-view consistency and complementarity while strengthening the synergistic interaction between view recovery and subspace representation. Extensive experimental results on seven datasets validate the effectiveness and robustness of the method. Overall, the paper is well-structured, with clear motivations and promising results, and some concerns require further exploration.

**Strengths:**

1).This paper proposes a method for aligning view recovery and tensor subspace learning using latent representations, which is well-structured.

2).The paper's research motivation is clear, the model design logic is clear, and it is easy to understand.

3).The experiments are detailed, and the method is compared with multiple baseline methods on multiple datasets at different missing rates and the experimental results show the method's good performance.

**Weaknesses:**

1).The background knowledge about tensors should be added with more details, which can help readers better understand the methods proposed in the paper.

2).The model involves alternating optimization of multiple variables, and the computational cost of tensors is high, lacking analysis of computational complexity.

3). The convergence of optimization process should be analyzed.

4). Does the tensor low-rank regularization preserve local manifold structures within each view, or mainly capture global correlations? Would adding a local structural constraint improve the model?

**Questions:**

See Weaknesses section.

---

> ### Author Response · Authors · 2025-11-21
> **Reply to Reviewer WN4G**
>
> *W1: The background knowledge about tensors should be added with more details, which can help readers better understand the methods proposed in the paper.*
>
> **Reply:** Thank you for your suggestion. Indeed, we utilize the low-rank tensor to explore the cross-view and cross-level correlation among multi-view. The background knowledge of the tensor and low-rank tensor should be provided, which helps readers understand better. In the revised version, we would add it.
>
> *W2: .The model involves alternating optimization of multiple variables, and the computational cost of tensors is high, lacking analysis of computational complexity.*
>
> **Reply:** The formulated objective function of the proposed ARSL-IMVC consists of multiple variables,for which is difficult to obtain closed-form solution. Thus, the alternating iteration strategy is used to update each variable. The computational complexity mainly relies on the updating rules of $\mathbf H$, $\mathbf P^v$, $\mathbf X_c^v$, $\mathbf Z^v$, $\mathbf Z$, $\mathbf E_1^v$, $\mathbf E_2^v$, $\mathbf E_H$, and $\mathcal J$. For variable $\mathbf H$, its updating rule mainly consists of matrix product and Sylvester equation,  requiring $O(k^{3} + n^{3}+nkd_v+n^2k+d_vk^2) \approx \mathcal O(n^3+nkd_v)$. For variable $\mathbf Z^v$,  its updating rule mainly consists of matrix product and inverse problem, requiring $\mathcal O(n^2d_v+n^3)$. Similarly, for variable $\mathbf Z$,  its updating rule mainly consists of matrix product and inverse problem, requiring $\mathcal O(n^2k+n^3) \approx \mathcal O(n^3)$. For variable $\mathbf X_c^v$,  its updating rule mainly consists of matrix product and inverse problem, requiring $\mathcal O(n^3+n^2d_v)$. For variable $\mathcal J$, its updating rule mainly consists of tensor singular value thresholding, requiring $O\left( (V+1)n^{2}\log n \right) + O\left( (V+1)^{2}n^{2} \right)$. For variable $\mathbf P^v$, its updating rule mainly consists of matrix product and matrix SVD, requiring $\mathcal O(nkd_v+k^2d_v)$. For variable $\mathbf E_1^v$, its updating rule mainly consists of matrix product and inverse problem, requiring $\mathcal O(n^3+n^2d_v)$. For variables $\mathbf E_2^v$ and $\mathbf E_H$, the updating rules mainly consist of matrix product and sparse thresholding operator, requiring $\mathcal O(n^2d_v)$ and $\mathcal O(n^2k)$. In summary, the ARSL-IMVC algorithm approximately requires $O( n^{3}+n^2d_v)$ complexity  in each iteration,  which is the same as most subspace representation-based IMVC baseline methods, such as HCP-IMSC and BWIC-TIMC. However, our method performs better. Further, the following running experiments (in seconds/iteration) demonstrate that the actual running costs of ARSL-IMVC are at a moderate level.
> | Methods   |   NGs  | BBCSport | 100leaves |   BDGP  |
> |-----------|-------:|---------:|----------:|--------:|
> | BSV       | 0.0146 | 0.8617   | 0.2263    | 1.2435  |
> | Concat    | 0.1168 | 1.7270   | 0.0711    | 0.2440  |
> | DAIMC     | 3.5722 | 9.3949   | 1.3958    | 1.0345  |
> | UEAF      | 0.6108 | 0.5109   | 1.3746    | 2.7080  |
> | IMSC-AGL  | 0.3718 | 0.3429   | 3.6214    | 8.5792  |
> | HCP-IMSC  | 0.3953 | 0.2910   | 3.4393    | 6.8946  |
> | BWIC-TIMC | 0.2921 | 0.2557   | 7.2357    | 8.9066  |
> | RMoGL     | 0.4818 | 0.5331   | 3.4361    | 12.4075 |
> | HCLS-CGL  | 0.2196 | 0.1280   | 1.6277    | 3.2471  |
> | Ours      | 0.5333 | 0.5174   | 2.7311    | 5.8518  |
>
> *W3: The convergence of optimization process should be analyzed.*
>
> **Reply:** This paper uses ADMM to solve subproblems for each variable. It is important to emphasize that the objective function involves non-convex problems such as tensor nuclear norm and HSIC regularization, and the various variables are tightly coupled. The strict theoretical convergence of ADMM optimization algorithm is difficult to analyze for this complex problem. This may be our future concern. Nevertheless, we provide an experimental convergence analysis by showing the variation of errors of equality constraints. See the revised version for more visual experiment results. It can be observed that the errors fluctuate and then steadily approach 0, verifying its stability.

---

> ### Author Response · Authors · 2025-11-21
> **Reply to Reviewer WN4G**
>
> *W4: Does the tensor low-rank regularization preserve local manifold structures within each view, or mainly capture global correlations? Would adding a local structural constraint improve the model?*
>
> **Reply:** In this paper, the primary goal of the tensor nuclear norm is to model global correlations across views. By approximating a low-rank tensor subspace, view-specific representations become more consistent in semantics and complementary in representation level, thus better fusing multi-view information. The local manifold structure within each view is primarily preserved through self-representation. As you suggested, the local constraint is a meaningful extension that can explicitly preserve the local adjacency structure, but it also increases the overall optimization time and space complexity. We will explore the feasibility of this extension in future work.

---

> > ### Comment · Reviewer_WN4G · 2025-11-26
> >
> > The author has addressed my concerns and I update my review score. I hope the authors can modify the submitted PDF according to the review comments. In addition, there is a minior concern. The solution of variable $\mathbf E_1^v$ in Equ (18) needs to be checked. The centralized matrix $\mathbf H$ in HSIC term and latent representation $\mathbf H$ are confused. The author needs further examination.

---

> > > ### Author Response · Authors · 2025-11-26
> > > **Reply to Reviewer WN4G**
> > >
> > > Thanks for increasing the score. The revised and rebuttal contents would be added in the final variant.

---

> ### Author Response · Authors · 2025-11-26
> **Reply to Reviewer WN4G**
>
> Thank you so much for your careful reading. According to your suggestions, we carefully reviewed and corrected it. Indeed, the centralized matrix $\mathbf H$ in HSIC term and latent representation $\mathbf H$ are confused. Here, in the revised version, the centralized matrix in HSIC is defined as $\mathbf{\tilde{H}}=\mathbf I_n - \frac{1}{n} \mathbf 1 \mathbf 1^T$. Thus, the optimal solution for varaible $\mathbf E_1^v$ should be as follows:
> $$
> \\mathbf{E}_1^{v} =(\\mathbf{X}_c^{v} - \\mathbf{P}^{v}\\mathbf{H} + {\\mathbf{Y}_1^{v}}/{\\mu}) (\\mathbf{I}+\\frac{2\\lambda_2}{\\mu(n-1)^2} \\sum\_{w=1; w \\neq v}^V \\mathbf{\\tilde{H}}\\mathbf{K}^{w} \\mathbf{\\tilde{H}})^{-1}$$
> Thank you.

---

### Official Review · Reviewer_K2t4 · 2025-10-28

**Soundness:** 3
**Presentation:** 3
**Contribution:** 3
**Rating:** 8
**Confidence:** 4

**Summary:**

The IMVC is an important topic and relies on discriminative representation learning. Most of existing imputation-based IMVC methods perform well, but they still face one crucial limitation, i.e., view recovery and subspace representation lack explicit alignment and collaborative interaction in exploring complementarity and consistency across multiple views. These issues are crucial for IMVC. This study proposes a novel ARSL-IMVC that aligns collaborative view recovery and tensorial Subspace Learning via latent representation. It is a simple yet effective method, and this is interesting design for IMVC. Extensive experimental results on seven datasets demonstrate the
powerful capacity of ARSL-IMVC in complex incomplete multi-view clustering tasks under various view missing scenarios. Overall, this paper has a clear and novel contribution to IMVC, and some concerns also need to be clarified.

**Strengths:**

1.The paper is well-written and easy to follow，ARSL-IMVC provides a new and cohesive framework that aligns consistency and complementarity across views.

2.The method analysis is relatively comprehensive, and the optimization steps are clearly explained. The experimental design is comprehensive, validating the performance of the method on seven datasets covering various missing rates, and comparing it with several representative baseline methods.The ablation and visualization analyses are well designed and strongly support the effectiveness of the proposed modules.

**Weaknesses:**

1.What is HW dataset? There is a lack of introduction to the statistical information of the used dataset. This is beneficial for readers.
Some experimental settings need to be further provided.

2. Some additional concerns can be found in **Questions**.

**Questions:**

1.When stacking view-shared and view-specific subspace representations into a tensor, does the optimization enforce any independence between shared and specific subspaces, making the features tend to be the same?

2.Besides the current averaging strategy, are there other fusion strategies explored to integrate shared and view-specific subspace representations? For example, learning an adaptive weight.

3.There is a lack of detailed analysis of the severe performance degradation of certain datasets under high missing rates. Further explanation and improvement strategies are recommended.

---

> ### Author Response · Authors · 2025-11-21
> **Reply to Reviewer K2t4**
>
> *W1: What is HW dataset? There is a lack of introduction to the statistical information of the used dataset. This is beneficial for readers. Some experimental settings need to be further provided.*
>
> **Reply:** We are sorry for these unclear descriptions. The "HW" is the abbreviation of Handwritten dataset, which consists of 2000 images of handwritten digits from 0-9. Here, each sample is described by 6 different features. Indeed, due to the space limitations, the statistics of the used datasets are not provided. In the revised version, the statistical information and description details of the used datasets are provided in the Appendix. In addition, more experimental settings are further provided.
>
> *Q1: When stacking view-shared and view-specific subspace representations into a tensor, does the optimization enforce any independence between shared and specific subspaces, making the features tend to be the same?*
>
> **Reply:** In the proposed ARSL-IMVC method, the view-shared and view-specific subspace representations are stacked into a unified low-rank tensor, which aims to explore the corss-view and cross-level high-order correlation and semantic interaction. The optimization of low-rank tensor woud weaken then noisy and redundant correlations. Specifically, it only promotes the consistent semantic information rather than completely consistent representations. In fact, the low-rank structure is imposed on rotated $n \times (V+1) \times n$ tensor $\mathcal Z$, where the classific tensor nuclear norm is defined as $$
> ||\mathcal{Z}||\_{\\circledast}
> = \\frac{1}{n} \\sum\_{k=1}^{n} ||\mathcal{Z}_f^{(k)}||\_{\ast}
> = \\frac{1}{n} \\sum\_{k=1}^{n} \\sum\_{i=1}^{V+1} \\mathcal{S}_f^{(k)}(i,i)
> $$
> where $\mathcal S_f$, where $\mathcal S_f$ is from the t-SVD of $\mathcal Z = \mathcal U* \mathcal S *{\mathcal V}^T$ in Fourier domain. In this specific case, the low-rank constraint mainly promotes the discriminative structure of $k$-th frontal slice of  $\mathcal Z$, exploring sample level association in a unified cross-view space. The high-order cross-view correlation is mainly due to singular value decomposition of unified tensor, which will realize cross-view information interaction. In addition, the subspace representation should satisfy self-representation reconstruction, which will preserve the diversity of original multi-view features. Thus, from feature representation level, the view-shared and view-specific representations would preserve more complementary and diverse information; but from semantic correlation level, they achieve the cross-view semantic interaction and promote the consistent semantics.
>
> *Q2: Besides the current averaging strategy, are there other fusion strategies explored to integrate shared and view-specific subspace representations? For example, learning an adaptive weight.*
>
> **Reply:** Indeed, this study utilizes a simple yet effective fusion strategy. The adaptive weight learning strategy is feasible. The ARSL-IMVC mainly focuses on aligning the view-recovery and subspace representation in exploring the complementarity and consistency across multiple views.  In fact, the ARSL-IMVC could implicitly adjust the role of various views. Within the framework of collaborative optimization, the shared structure naturally enhances the weights of reliable views. In addition, when the model itself is already complex, weight learning introduces additional variables, increasing optimization complexity and potentially amplifying the impact of noisy views. Ultimately, we chose the more stable and concise average fusion strategy. In the future, we would consider parameter-free fusion strategy.
>
> *Q3: There is a lack of detailed analysis of the severe performance degradation of certain datasets under high missing rates. Further explanation and improvement strategies are recommended.*
>
> **Reply:** When the missing data rate is extremely high, the performance of a few datasets drops significantly across all methods because these datasets exhibit high heterogeneity in view distribution and weak cross-view complementarity. The drastic reduction in observable data increases the difficulty of inferring accurate missing views from a very small number of samples, consequently reducing the reliability of the learned subspace representation. Despite this, our method still maintains good performance, demonstrating the model's effectiveness under high missing data rates. The diffusion generation model may be beneficial for missing data recovery, mainly because it uses forward and reverse processes to simulate data missing for recovery, and can fit the distribution of multi-view data to improve the quality of recovery. We would further disscuss this issue in the revised version.

---

### Official Review · Reviewer_seS7 · 2025-10-30

**Soundness:** 3
**Presentation:** 3
**Contribution:** 3
**Rating:** 6
**Confidence:** 5

**Summary:**

This work addresses incomplete multi-view clustering by introducing an aligned representation and subspace learning model. Unlike conventional imputation-based methods, ARSL-IMVC employs a shared latent representation to guide view recovery and tensor subspace construction simultaneously, thus promoting collaborative information exchange across views. The framework is novel and effective, enabling the model to capture both global coherence and view-level diversity. Comprehensive experiments across multiple datasets demonstrate the model's robustness and superior clustering performance under different missing view scenarios. The idea is novel, clear, and provides a meaningful advancement for incomplete multi-view clustering.

**Strengths:**

1. The paper proposes a novel framework for joint view recovery and tensor subspace learning. It explicitly aligns the two modules through a shared latent representation, which is relatively uncommon in existing IMVC methods.

2. The experimental design is comprehensive, covering various missing rates and including comparisons with multiple baseline methods. The results are superior and clearly demonstrate the effectiveness of each module.

**Weaknesses:**

1. Lack of introduction to datasets and baseline methods.

2. While results are averaged over 10 runs, reporting the standard deviation would make the experimental conclusions more statistically convincing.

3. Two main differences as induced in this paper need to be more detailed. And the authors are suggested to give more explanations related to some key words, e.g., "limited structural fidelity", "insufficient diversity and consistency reshaping", etc. This makes it easier for readers to understand these opinions. And some empirical or theoretical evidence is more favoured.

4. Notations are not consistent. Please see my questions.

5. The motivations of some equation formulations are not clear. Please see my questions.

6. Complexity analysis is lacking.

**Questions:**

Apart from the above weaknesses, I still have some questions:

1. The semantic alignment between view recovery and subspace representation is new, its innovation needs further clarification, especially the difference from existing methods.

2. Exploring the criteria of cross-view consistency and complementarity has been widely used in multi-view learning. What is the contribution of this study in this regard?

3. If the latent representation has already been trained to capture shared semantics, does the introduction of a low-rank constraint on the subspace representation tensor result in redundant regularisation?

4. The HSIC regularisation term can use different kernel functions. Why does ARSL-IMVC choose to use the inner product kernel?

5. The latent representation H and the centralised matrix H use the same notation.

6. The authors introduced a view-specific orthogonal matrix $P^{(v)}$ to recover the view-specific features. I have a concern about whether $(P^{(v)})^\top P^{(v)} = I$ can still hold when the dimension of view-specific features is lower than $k$.

7. The choice of HSIC: why this paper chooses HSIC as a diversity measurement here rather than some others, e.g., Gromov-Wasserstein distance.

8. What are the global semantics and local cluster structures in Eq. (4) ?

9. This paper utilises a rotation operator to transfer a $N\times N\times (V+1)$ tensor to a $ N\times (V+1) \times N$  tensor, making the results of the tensor low rank constraint sensitive to the orders of samples arranged in the input features. How does this paper address this issue?

10. The datasets used in this paper are small-scale.

The scores of this paper will be raised after addressing the above weaknesses and questions.

---

> ### Author Response · Authors · 2025-11-21
> **Reply to Reviewer seS7**
>
> Thank you very much for your suggestions. We now provide further clarification on these questions:
>
> *W1: Lack of introduction to datasets and baseline methods.*
>
> **Reply:** Here, we would provide the statistics of the used datasets in the following table, and more detailed descriptions be added in the Appendix of revised version.
>
> | Dataset     | Samples | Classes | Views | Feature              |
> |------------|---------|---------|-------|----------------------|
> | NGs        | 500     | 5       | 3     | 2000×2000×2000       |
> | 100leaves  | 1600    | 100     | 3     | 64×64×64             |
> | BDGP       | 2500    | 5       | 2     | 1000×500             |
> | Handwritten| 2000    | 10      | 6     | 76×216×64×24×47×7    |
> | BBCSport   | 544     | 5       | 2     | 3183×3203            |
> | Yale       | 165     | 15      | 3     | 4096×3304×6750       |
> | Scene-15   | 4485    | 15      | 3     | 20×59×40             |
>
> For the compared methods, partial methods are intoducted in detail in the Related Work section. Now, we provide summary introduction to competitors for intuitive understanding of readers. And, we would add the introduction of baseline methods in the revised version. Here, we copy them as follows.
>
> BSV: It is a baseline method that fills incomplete views by averaging existing instances and performs spectral clustering on each single view independently. The best results from various views are reported.
>
> Concat: It concatenates features from all views and applies spectral clustering in the merged feature space.
>
> DAIMC:  It is a matrix factorization-based IMVC method, which learns a shared latent representation via weighted non-negative matrix factorization across views.
>
> IMSC-AGL:  It is a classic incomplete multi-view subspace clustering method, which performs graph learning and similarity alignment under missing data.
>
> UEAF: It simultaneously performs view reconstruction, local structure preservation, and adaptive view-weighting within a unified learning framework.
>
> HCP-IMSC: It is a tensor-based IMVC method, which uses hyper-Laplacian regularization to preserve high-order correlations and recover missing views.
>
> HCLS-CGL: It captures group-wise structural information by constructing a confidence neighbor graph that models nearest-neighbor probabilities between samples.
>
> BWIC-TIMC:  It leverages tensor constraints to jointly capture intra-view structure and inter-view consistency, and adopts an adaptive fusion strategy to learn a consensus representation for clustering.
>
> RMoGL: It improves noise resistance by decoupling clean and noisy graphs and adaptively learning a high-order consensus structure from recovered multi-view data.

---

> ### Author Response · Authors · 2025-11-21
> **Reply to Reviewer seS7**
>
> *Q1: The semantic alignment between view recovery and subspace representation is new, its innovation needs further clarification, especially the difference from existing methods.*
>
> **Reply:** Most existing IMVC methods treat view restoration and subspace learning as a weakly coupled joint optimization. However, we establish a bidirectional feedback channel by sharing a latent representation $\mathbf H$. This latent representation is used to reconstruct each view, and also correlates with the self-representations of each view through low-rank tensor constraints. This aligns the restored views and the clustering structure within the same semantic space, achieving a deep, semantic collaboration, rather than simply two tasks sharing a single objective function.
>
> *Q2: Exploring the criteria of cross-view consistency and complementarity has been widely used in multi-view learning. What is the contribution of this study in this regard?*
>
> **Reply:** This paper does not simply weight cross-view consistency and complementarity metrics, but rather considers both within the same alignment space. First, it maps each view to the same alignment space by sharing a latent representation H, and applies tensor low-rank constraints to this space to characterize the shared global structure across views. Based on this, it introduces HSIC diversity regularization to constrain the unique components of each view, encouraging them to provide non-redundant complementary information, thus achieving joint learning of consistency and complementarity.
>
> *Q3: If the latent representation has already been trained to capture shared semantics, does the introduction of a low-rank constraint on the subspace representation tensor result in redundant regularisation?*
>
> **Reply:** The Latent representation enforces consistency at the feature level, learning a shared representation, while tensor low-rank constraints are imposed on the self-representation coefficients of each view, focusing more on the sample level and encouraging consistency in the similarity structure of different views within the subspace. In incomplete scenarios, even if the latent representation has captured the main semantics, the recovered data may still contain noise or local distortions. Introducing the low-rank prior of the tensor quantum space can align the relational structure; therefore, it is complementary to latent representation rather than a simple repetition.
>
> *Q4: The HSIC regularisation term can use different kernel functions. Why does ARSL-IMVC choose to use the inner product kernel?*
>
> **Reply:** We chose the inner product kernel primarily for its computational efficiency and inherent properties. Our method employs the assumption of subspace self-representation, and the linear kernel aligns better with this assumption, avoiding the introduction of additional nonlinear assumptions into the HSIC regularization term. Furthermore, the linear kernel is simple in form, requires no additional kernel parameters, and has low computational complexity.
>
> *Q5: The latent representation H and the centralised matrix $\mathbf H$ use the same notation.*
>
> **Reply:** Thank you for pointing out this inaccuracy. In the definition of HSIC, the centralized matrix is ​​generally represented by $\mathbf H$, but we have already used $\mathbf H$ as the latent representation in our paper, which may mislead readers. We will correct the conflicting notation in the paper.
>
> *Q6:The authors introduced a view-specific orthogonal matrix $\mathbf P^{v}$ to recover the view-specific features. I have a concern about whether $(\mathbf P^{v})^{\top}\mathbf P^{v}=\mathbf I$ can still hold when the dimension of view-specific features is lower than.*
>
> **Reply:** In the HW dataset, one view has a feature dimension of 7, while the best setting in the experiments was $k=20$, under which the model's performance remained unaffected. In this case, latent representation has some "redundancy," but this “redundancy” is highly correlated with the effective columns, so it has little impact on the objective function, our method remains effective.
>
> *Q7: The choice of HSIC: why this paper chooses HSIC as a diversity measurement here rather than some others, e.g., Gromov-Wasserstein distance.*
>
> **Reply:** The choice of HSIC as a measure of diversity among views is primarily based on considerations of computational efficiency and optimization feasibility. While the Gromov–Wasserstein distance is highly expressive in measuring structural differences between distributions, it requires solving an optimal transport problem, resulting in high computational complexity and significantly increasing time costs within an iterative optimization framework.

---

> ### Author Response · Authors · 2025-11-21
> **Reply to Reviewer seS7**
>
> *Q8: What are the global semantics and local cluster structures in Eq.(4)?*
>
> **Reply:** In this paper, "global semantics" mainly refers to the subspace representation learned by the shared latent representation $\mathbf H$, which represents a consistent consensus on sample relations across views; the corresponding terms model this global semantic structure by constraining the consistency between $\mathbf H$ and the recovered view and the self-representation tensor. "local cluster structures" refers to the subspace self-representation learned from each recovered specific view, which captures the local neighborhood structure of samples in each view. Eq.(4) establishes cross-view global semantic alignment through $\mathbf H$ on the one hand, and maintains a reasonable local cluster structure in each view on the other.
>
> *Q9: This paper utilises a rotation operator to transfer a $N\times N\times (V+1)$ tensor to a $ N\times (V+1) \times N$ tensor, making the results of the tensor low rank constraint sensitive to the orders of samples arranged in the input features. How does this paper address this issue?*
>
> **Reply:** Tensor rotation is used to perform $t$-SVT and tensor nuclear norm along a specific modulus. It only performs a fixed axial rearrangement between the three moduli of the tensor, without shuffling the samples within each dimension or changing the relative positions between the samples. If all relevant tensors are rearranged in the same way, the Frobenius norm and tensor nuclear norm values ​​involved in the objective function remain unchanged. Therefore, the tensor rotation here is merely a numerical implementation trick and does not introduce additional sample order sensitivity.
>
> *Q10: The datasets used in this paper are small-scale.*
>
> **Reply:** Since computational complexity is related to the number of samples, extremely large datasets may cause a dramatic increase in the algorithm's runtime. In our future work, we will focus on incomplete multi-view clustering of large datasets to improve the scalability of our methods on large datasets.

---

> ### Comment · Reviewer_seS7 · 2025-11-21
> **Response to authors**
>
> The authors present two points to explain the issues of "limited structural fidelity" and "insufficient diversity and consistency reshaping." However, I believe these should not be labelled as facts unless the authors can provide empirical or theoretical evidence to support these assertions.

---

> ### Comment · Reviewer_seS7 · 2025-11-21
> **Response to the authors' feedback on Q2 to Q10**
>
> 1. Related to Q6. The authors did not clarify whether $(\mathbf P^{v})^{\top}\mathbf P^{v}=\mathbf I$ can still hold when the dimension of view-specific features is lower than $k$.
>
> 2. Related to Q7. In comparison to HSIC, what additional computational complexity is introduced by using the Gromov-Wasserstein distance? Furthermore, will clustering performance improve or not when HSIC is replaced with the Gromov-Wasserstein distance?
>
> 3.  Related to Q8. I suggest that the authors replace "global semantics" and "local cluster structures" with "view-shared subspace representation" and "view-specific subspace representation" for greater clarity.
>
> 4. Related to Q9. Regarding tensor constraints, the order of samples or views in the tensor can significantly impact the Fast Fourier Transform (FFT) procedure during tensor production. The Discrete Fourier Transform (DFT) plays a crucial role in tensor-tensor products. The DFT on $ \mathbf{x} \in \mathbb{R}^n $ is given by $\bar{x} = \mathbf{P}_n \mathbf{x}$
>
> $$
> \mathbf{P}_n = \begin{bmatrix}
> 1 & 1 & 1 & \cdots & 1 \\
> 1 & \omega & \omega^2 & \cdots & \omega^{n-1} \\
> \vdots & \vdots & \vdots & \ddots & \vdots \\
> 1 & \omega^{n-1} & \omega^{2(n-1)} & \cdots & \omega^{(n-1)(n-1)}
> \end{bmatrix} \in \mathbb{C}^{n \times n},
> $$
> where $\omega_n = e^{-2\pi i / n} $. From this procedure, the FFT results depend on the order in which the samples are arranged in the matrix. How does this paper address this issue?
>
> 5. Related to Q10. Please provide some results on medium-sized MVC datasets.

---

> ### Author Response · Authors · 2025-11-23
> **Reply to Reviewer seS7 for W2**
>
> *W2: While results are averaged over 10 runs, reporting the standard deviation would make the experimental conclusions more statistically convincing.*
>
> **Reply:** In our experiments, we ran each method 10 times independently and calculated the corresponding standard deviation. Here, according to your suggestions, we show the average vaules and standard deviation on BBCSport, HW, and BDGP dataset. The experimental results of representative competitors and our ARSL-IMVC are shown as follows, and the complete results of all methods are provided in the Appendix of revised version. From the following results, the experimental results are relatively stable.
>
> | Dataset | p   | Metrics | HCP-IMSC        | RMoGL           | Ours           |
> |---------|-----|---------|-----------------|-----------------|----------------|
> | BBCSport     | 0.1 | ACC     | 91.91±0.00      | 89.19±0.00      | 96.51±0.00     |
> |         |     | NMI     | 79.84±0.00      | 83.18±0.00      | 89.77±0.00     |
> |         |     | Purity  | 91.91±0.00      | 88.79±0.00      | 96.51±0.00     |
> | BBCSport     | 0.3 | ACC     | 89.15±0.00      | 78.49±0.00      | 94.85±0.00     |
> |         |     | NMI     | 75.47±0.00      | 68.86±0.00      | 84.95±0.00     |
> |         |     | Purity  | 89.15±0.00      | 81.80±0.00      | 94.85±0.00     |
> | BBCSport     | 0.5 | ACC     | 86.05±0.00      | 76.47±0.00      | 88.97±0.00     |
> |         |     | NMI     | 72.37±0.00      | 61.76±0.00      | 71.32±0.00     |
> |         |     | Purity  | 86.05±0.00      | 79.60±0.00      | 88.97±0.00     |
> | HW      | 0.1 | ACC     | 79.80±0.04      | 76.73±0.36      | 96.90±0.00     |
> |         |     | NMI     | 75.73±0.05      | 74.61±0.17      | 92.77±0.00     |
> |         |     | Purity  | 80.05±0.04      | 76.73±0.36      | 96.90±0.00     |
> | HW      | 0.3 | ACC     | 75.35±0.05      | 64.19±0.03      | 92.46±0.03     |
> |         |     | NMI     | 69.11±0.17      | 62.25±0.04      | 84.16±0.04     |
> |         |     | Purity  | 76.50±0.05      | 65.03±0.03      | 92.46±0.03     |
> | HW      | 0.5 | ACC     | 70.80±0.02      | 59.20±0.21      | 89.03±0.09     |
> |         |     | NMI     | 60.47±0.01      | 54.74±0.19      | 77.97±0.12     |
> |         |     | Purity  | 71.30±0.02      | 59.35±0.21      | 89.03±0.16     |
> | BDGP    | 0.1 | ACC     | 21.08±0.12      | 45.94±0.03      | 56.07±0.02     |
> |         |     | NMI     | 25.26±0.18      | 23.48±0.02      | 35.22±0.08     |
> |         |     | Purity  | 19.52±0.12      | 46.66±0.02      | 56.07±0.01     |
> | BDGP    | 0.3 | ACC     | 23.93±0.05      | 42.66±0.03      | 50.58±0.03     |
> |         |     | NMI     | 29.06±0.07      | 20.30±0.08      | 31.59±0.00     |
> |         |     | Purity  | 22.21±0.05      | 43.60±0.03      | 52.07±0.01     |
> | BDGP    | 0.5 | ACC     | 20.46±0.00      | 31.68±0.00      | 49.21±0.07     |
> |         |     | NMI     | 24.84±0.00      | 6.77±0.00       | 32.16±0.05     |
> |         |     | Purity  | 19.00±0.00      | 32.68±0.00      | 50.21±0.09     |

---

> ### Author Response · Authors · 2025-11-25
> **Reply to  Reviewer seS7**
>
> **Reply:** Here, we provide detailed explanations for them. In fact, the "limited structural fidelity" and "insufficient diversity and consistency reshaping" are interrelated. In this study, the structural fidelity mainly refers to the fact that multi-view data itself contains cross-view representation diversity and semantic consistency. Thus, this study claims that most existing  imputation-based IMVC methods do not explicitly model cross-view consistency and diversity in view recovery, resulting in limited structural fidelity of recovered multi-view data. For ease of understanding, we theoretically provide comparison and analysis with several representative methods, including RMoGL, HCP-IMSC, IMVC-HG.
>
> For RMoGL, the view recovery strategy is $\mathbf X^{(v)}+\mathbf B^{(v)}\mathbf W^{(v)}$, where $\mathbf X^{(v)}$ is the incomplete data matrix of  the $v$-th view and the missing samples are imputed with zeros, $\mathbf B^{(v)}\mathbf W^{(v)}$ is used to recover the missing instances of the original data in the $v$-th view. Then, the recovered data $\mathbf X^{(v)}+\mathbf B^{(v)}\mathbf W^{(v)}$ is used to learn subspace representation $\mathbf Z^{(v)}$ by self-representation and exploring the cross-view correlations on $\mathbf Z^{(v)}$. Obviously, the RMoGL indirectly affects view recovery through the correlation exploration on $\mathbf Z^{(v)}$, which does not explicitly models the complex inherent correlations (representation diversity and semantic consistency) across multiple views in the view recovery. In summary, the RMoGL struggles to explore cross-view complementarity and consistency in both view recovery and subspace representation learning with an aligned manner.
>
> For HCP-IMSC and IMVC-HG, they utilize the following view recovery strategy, i.e., $\mathbf X^{(v)}=\mathbf X_o^{(v)}\mathbf P_o^{(v)}+\mathbf X_u^{(v)}\mathbf P_u^{(v)}$, where $\mathbf X_o^{(v)}$ is the feature matrix of observed samples in $v$-th view, $\mathbf P_o^{(v)}$ and $\mathbf P_u^{(v)}$ are the indicator matrices which project the observed and missing samples into the corresponding indicators in the complete data, and $\mathbf X_u^{(v)}$ is the learnable missing feature matrix. Similarly, the recovered complete data $\mathbf X^{(v)}$ is utilized to learn subspace representation $\mathbf Z^{(v)}$ by self-representation and exploring the cross-view correlations on $\mathbf Z^{(v)}$ with various regularizers. Hence, in fact, they model complex cross-view correlations on subspace representations and also indirectly react on feature recovery, rather than explicit modeling.
>
> Different from them, the proposed ARSL-IMVC explicitly explores the cross-view complementarity and consistency in both view recovery and subspace representation learning, and aligns them by shared latent feature representation. The view recovery in ARSL-IMVC considers the cross-view correlations (diversity and consistency), i.e., $\mathbf X^{v}\mathbf W^{v}=\mathbf P^{v}\mathbf H+\mathbf E_1^{v}$ with diversity iterm $HSIC(\mathbf E_1^v, \mathbf E_2^w)$, thus the subspace representations have clear structural fidelity. Specifically, the subspace representations could accurately reflect the cross-view correlations and clustering structure.
>
> In addition to the above theoretical analysis, we also provide simple experimental verification. Here, we design the evaluation indicators for representation diversity and semantic consistency. For representation diversity, the cosine similarity is used to measure the diversity of recovered views. The Normalized Mutual Information (NMI) is utilized to measure the label similarity among any recovered views, which reflects the semantic consistency. The experimental results of representative methods are shown as follows. As shown in the following results, the cosine similarity of the proposed ARSL-IMVC is relatively small in most cases, i.e., the representation diversity is strong. In addition, the NMI of the proposed ARSL-IMVC is relatively large in most cases, i.e., the semantic consistency is strong. On the whole, the proposed ARSL-IMVC method achieves a good balance between the representation diversity and semantic consistency for recovered views.
>
> | Cosine Similarity   | HCP-IMSC  | IMVC-HG  | RMoGL  | Our    |
> |-----------|--------|--------|--------|--------|
> | NGs       | 0.0364 | 0.0028 | 0.0033 | 0.0035 |
> | Yale      | 0.0867 | 0.0089 | 0.0464 | 0.0197 |
> | 100leaves       | 0.1169 | 0.0009 | 0.0068 | 0.0127 |
> | BBCSport  | 0.0635 | 0.0026 | 0.0038 | 0.0023 |
> | HW        | 0.1865 | 0.0007 | 0.0094 | 0.0336 |
> | BDGP      | 0.2152 | 0.0258 | 0.0029 | 0.0027 |
>
>
> | NMI    | HCP-IMSC  | IMVC-HG  | RMoGL  | Our    |
> |-----------|------|------|-------|------|
> | NGs       | 0.57 | 0.08 | 0.37  | 0.64 |
> | Yale      | 0.95 | 0.17 | 0.62  | 0.65 |
> | 100leaves     | 0.35 | 0.23 | 0.58  | 0.90 |
> | BBCSport  | 0.81 | 0.06 | 0.27  | 0.12 |
> | HW        | 0.76 | 0.03 | 0.49  | 0.96 |
> | BDGP      | 0.30 | 0.09 | 0.30  | 0.31 |

---

> ### Author Response · Authors · 2025-11-25
> **Reply to Reviewer seS7 on W3**
>
> **Reply:** Here, we provide detailed explanations for them. In fact, the "limited structural fidelity"and "insufficient diversity and consistency reshaping" are interrelated. In this study, the structural fidelity mainly refers to that multi-view data itself contains cross-view representation diversity and semantic consistency. Thus, this study claims that the most existing  imputation-based IMVC methods do not explicitly model cross-view consistency and diversity in view recovery, resulting in limited structural fidelity of recovered multi-view data. For ease of understanding, we theoretically provide comparison and analysis with several representative methods, including RMoGL, HCP-IMSC, IMVC-HG.
>
> For RMoGL, the view recovery strategy is $\mathbf X^{(v)}+\mathbf B^{(v)}\mathbf W^{(v)}$, where $\mathbf X^{(v)}$ is the incomplete data matrix of  the $v$-th view and the missing samples are imputed with zeros, $\mathbf B^{(v)}\mathbf W^{(v)}$ is used to recover the missing instances of the original data in the $v$-th view. Then, the recovered data $\mathbf X^{(v)}+\mathbf B^{(v)}\mathbf W^{(v)}$ is used to learn subspace representation $\mathbf Z^{(v)}$ by self-representation and exploring the cross-view correlations on $\mathbf Z^{(v)}$. Obviously, the RMoGL indirectly affects view recovery through the correlation exploration on $\mathbf Z^{(v)}$, which not explicitly models the complex inherent correlations (representation diversity and semantic consistency) across multiple views in the view recovery. In summary, the RMoGL struggles to explore cross-view complementarity and consistency in both view recovery and subspace representation learning with an aligned manner.
>
> For HCP-IMSC and IMVC-HG, they utilize the following view recovery strategy, i.e., $\mathbf X^{(v)}=\mathbf X_o^{(v)}\mathbf P_o^{(v)}+\mathbf X_u^{(v)}\mathbf P_u^{(v)}$, where $\mathbf X_o^{(v)}$ is the feature matrix of observed samples in $v$-th view, $\mathbf P_o^{(v)}$ and $\mathbf P_u^{(v)}$ are the indicator matrices which project the observed and missing samples into corresponding indicator in complete data, and $\mathbf X_u^{(v)}$ is the learnable missing feature matrix. Similarly, the recovered complete data $\mathbf X^{(v)}$ is utilized to learn subspace representation $\mathbf Z^{(v)}$ by self-representation and exploring the cross-view correlations on $\mathbf Z^{(v)}$ with various regularizers. Hence, in fact, they model complex cross-view correlations on subspace representations and also indirectly reacts on feature recovery, rather than explicit modeling.
>
> Different from them, the proposed ARSL-IMVC explicitly explores the cross-view complementarity and consistency in both view recovery and subspace representation learning, and aligns them by shared latent feature presentation. The view recovery in ARSL-IMVC considers the cross-view correlations (diversity and consistency), i.e., $\mathbf X^{v}\mathbf W^{v}=\mathbf P^{v}\mathbf H+\mathbf E_1^{v}$ with diversity iterm $HSIC(\mathbf E_1^v, \mathbf E_2^w)$, thus the subspace representations have clear structural fidelity. Specifically, the subspace representations could accurately reflect the cross-view correlations and clustering structure.
>
> In addition to the above theoretical analysis, we also provide simple experimental verification. Here, we design the evaluation indicators for representation diversity and semantic consistency. For representation diversity, the cosine similarity is used to measure the diversity of recovered views. The Normalized Mutual Information (NMI) is utilized to measure the label similarity among any recovered views, which reflects the semantic consistency. The experimental results of representative methods are shown as follows. As shown in the following results, the cosine similarity of the proposed ARSL-IMVC is relatively small in most cases, i.e., the representation diversity is strong. In addition, the NMI the proposed ARSL-IMVC is relatively large in most cases, i.e., the semantic consistency is strong. On the whole, the proposed ARSL-IMVC method achieves a good balance between the representation diversity and semantic consistency for recovered views.
>
> | Cosine Similarity   | HCP-IMSC  | IMVC-HG  | RMoGL  | Our    |
> |-----------|--------|--------|--------|--------|
> | NGs       | 0.0364 | 0.0028 | 0.0033 | 0.0035 |
> | Yale      | 0.0867 | 0.0089 | 0.0464 | 0.0197 |
> | 100leaves       | 0.1169 | 0.0009 | 0.0068 | 0.0127 |
> | BBCSport  | 0.0635 | 0.0026 | 0.0038 | 0.0023 |
> | HW        | 0.1865 | 0.0007 | 0.0094 | 0.0336 |
> | BDGP      | 0.2152 | 0.0258 | 0.0029 | 0.0027 |
>
>
> | NMI    | HCP-IMSC  | IMVC-HG  | RMoGL  | Our    |
> |-----------|------|------|-------|------|
> | NGs       | 0.57 | 0.08 | 0.37  | 0.64 |
> | Yale      | 0.95 | 0.17 | 0.62  | 0.65 |
> | 100leaves     | 0.35 | 0.23 | 0.58  | 0.90 |
> | BBCSport  | 0.81 | 0.06 | 0.27  | 0.12 |
> | HW        | 0.76 | 0.03 | 0.49  | 0.96 |
> | BDGP      | 0.30 | 0.09 | 0.30  | 0.31 |

---

> ### Author Response · Authors · 2025-11-25
> **Reply to Reviewer seS7 on W6**
>
> *W6: Complexity analysis is lacking.*
>
> **Reply:** The formulated objective function of the proposed ARSL-IMVC consists of multiple variables,for which is difficult to obtain closed-form solution. Thus, the alternating iteration strategy is used to update each variable. The computational complexity mainly relies on the updating rules of $\mathbf H$, $\mathbf P^v$, $\mathbf X_c^v$, $\mathbf Z^v$, $\mathbf Z$, $\mathbf E_1^v$, $\mathbf E_2^v$, $\mathbf E_H$, and $\mathcal J$. For variable $\mathbf H$, its updating rule mainly consists of matrix product and Sylvester equation,  requiring $O(k^{3} + n^{3}+nkd_v+n^2k+d_vk^2) \approx \mathcal O(n^3+nkd_v)$. For variable $\mathbf Z^v$,  its updating rule mainly consists of matrix product and inverse problem, requiring $\mathcal O(n^2d_v+n^3)$. Similarly, for variable $\mathbf Z$,  its updating rule mainly consists of matrix product and inverse problem, requiring $\mathcal O(n^2k+n^3) \approx \mathcal O(n^3)$. For variable $\mathbf X_c^v$,  its updating rule mainly consists of matrix product and inverse problem, requiring $\mathcal O(n^3+n^2d_v)$. For variable $\mathcal J$, its updating rule mainly consists of tensor singular value thresholding, requiring $O\left( (V+1)n^{2}\log n \right) + O\left( (V+1)^{2}n^{2} \right)$. For variable $\mathbf P^v$, its updating rule mainly consists of matrix product and matrix SVD, requiring $\mathcal O(nkd_v+k^2d_v)$. For variable $\mathbf E_1^v$, its updating rule mainly consists of matrix product and inverse problem, requiring $\mathcal O(n^3+n^2d_v)$. For variables $\mathbf E_2^v$ and $\mathbf E_H$, the updating rules mainly consist of matrix product and sparse thresholding operator, requiring $\mathcal O(n^2d_v)$ and $\mathcal O(n^2k)$. In summary, the ARSL-IMVC algorithm approximately requires $O( n^{3}+n^2d_v)$ complexity  in each iteration,  which is the same as most subspace representation-based IMVC baseline methods, such as HCP-IMSC and BWIC-TIMC. However, our method performs better.

---

> ### Author Response · Authors · 2025-11-25
> **Reply to  Reviewer seS7 on Q6, Q7, Q8, Q9**
>
> Q6: Here, we will provide the detailed explanations for this issue. First, in most cases, the original multi-view data is high-dimensional, and $k \leq {d_v}$ is usually satisfied, i.e., $(\mathbf P^v)^T\mathbf P^v = \mathbf I$ holds. Second, when $d_v \leq k$, the solution in Eq (8) does not satisfy the orthogonal constraint $(\mathbf P^v)^T\mathbf P^v = \mathbf I$. The utilized optimization algorithm is an iterative algorithm, which essentially obtains the approximate solution of the ideal optimal solution. Therefore, when $d_v \leq k$, we directly use the solution in Eq (8) as the approximate solution. From the experimental results, its effect on the performance is small. Although, according to your suggestion, we also tried to solve this issue. Specifically, the orthogonal constraint is transformed into orthogonal regularization $\lambda\|\|(\mathbf P^v)^T\mathbf P^v = \mathbf I\|\|_F^2$, which is then introduced into the objective function. By optimizing this regularization term, $\mathbf P^v$ will be encouraged to maintain orthogonality as much as possible during the optimization process.
>
> Q7: We will further provide detailed explanations from the following aspects:
>
> 1)  The HSIC term penalties statistical correlation to measure similarity, while the Gromov-Wasserstein distance mainly measures the similarity of the internal structure space. From the perspective of pursuing diversity, statistical correlation is a more appropriate metric. If semantic structure similarity is promoted, the Gromov-Wasserstein distance metric be more appropriate.
>
> 2) For optimization process, the maximization of Gromov-Wasserstein distance between $\mathbf E_1^v$ and $\mathbf E_1^w$ may increase the computational complexity. The empirical Gromov-Wasserstein distance among $\mathbf E_1^v$ and $\mathbf E_1^w$ is defined as
>
> $d_{gw}(\mathbf E_1^v,\mathbf E_1^w) = \max_{\mathbf T \in \Pi(\hat{\rho}_v, \hat{\rho}_w)} \, \text{tr}\left(C_v^T \mathbf T^T C_v \mathbf T \right)$
>
> where $C_v$ and $C_w$ are two relation matrices constructed by the samples, and each element ${C_v}(ij)$ indicates the relation between $\mathbf E_1^v(:,i)$ and $\mathbf E_1^v(:,j)$, and ${C_w}(ij)$ is defined in $\mathbf E_1^w$ in the same way. For example, ${C_v}(ij)=\|\|\mathbf E_1^v(:,i)-\mathbf E_1^v(:,j)\|\|_2^2$ is a representative relation way. From the formulation of empirical Gromov-Wasserstein distance, it should solve the above optimization problem to obtain distance, rather than direct calculation. Thus, if we use the maximization of Gromov-Wasserstein distance as the diversity constraint, the updating process of variable $\mathbf E_1^v$ needs to calculate the transport matrix $\mathbf T$ additionally.  For HSIC term, it is a direct function of variables $\mathbf E_1^v$ and $\mathbf E_1^w$, which can directly measure independence and facilitate the optimization process. Therefore, using HSIC will not increase the computational cost too much.
>
> 3) The empirical Gromov-Wasserstein distance is complex. We select its simplified version to test performance. The simplified Gromov-Wasserstein distance is $d_{sgw}(\mathbf E_1^v,\mathbf E_1^w) = \frac{1}{n}\|\|C_v-C_w\|\|_F^2$, which is used to replace the HSIC term. The experimental results are as follows. We can observe that ARSL-IMVC with HSIC has performance advantages over the ablation version ARSL-IMVC with simplified Gromov-Wasserstein distance metric.
>
> | Dataset   | HSIC |         |          |  $ d_{sgw}(\mathbf E_1^v,\mathbf E_1^w)$   |          |   |
> |-----------|-----------|--------|--------|----------------------|--------------|-----------|
> | **p**     | **0.1** | **0.3** | **0.5** | **0.1** | **0.3** | **0.5** |
> | Yale      | 86.06   | 89.70   | 82.42   | 77.21   | 88.48   | 71.29   |
> | NGs       | 96.20   | 95.20   | 93.00   | 90.68   | 84.70   | 85.10   |
> | 100leaves | 92.50   | 90.64   | 88.41   | 85.23   | 85.24   | 84.25   |
> | BDGP      | 56.07   | 50.58   | 49.21   | 52.96   | 40.42   | 39.03   |
> | BBCSport  | 96.51   | 94.85   | 88.97   | 81.70   | 84.74   | 80.99   |
>
> Q8: Thank you. We would revise them in the revised version.
>
> Q9: This is a valuable question. Indeed, the FFT results depend on the order in which the samples are arranged in the matrix. This is not the focus of this paper, we did not give special consideration to this issue. Through the actual experimental results, our method is effective and stable. In the future, we will focus on this. Thank you !

---

> ### Author Response · Authors · 2025-11-25
> **Reply to Reviewer seS7 on Q10**
>
> Q10: Here, we test the proposed ARSL-IMVC on the medium-sized MVC dataset. The Handwritten digits dataset with $\mathbf{10000}$ samples is utilized, two views are collected from various resources: MNIST and USPS. The experimental results in $0.1$ missing ratio are shown as follows. The ARSL-IMVC still achieves the optimal clustering performance. In fact, the computational costs of these methods are high, and we need to design economical methods with strong scalability in the future.
>
> | Methods | DAIMC | UEAF | IMSC-AGL | HCLS-IMSC | HCP-IMSC |  Our |
> |---------|-------|-------|----------|-----------|----------|------|
> | ACC     | 67.58 | 85.38 | 76.32 | 98.29   | 89.56    |  99.00 |
> | NMI     | 64.25 | 73.56 | 77.40  | 95.30   | 89.52    | 96.97 |
> |Purity   | 69.61 | 85.38 | 77.89  |  98.29  | 88.40 |   99.00 |

---

> > ### Comment · Reviewer_seS7 · 2025-11-28
> > **Reply to Authors**
> >
> > Thanks to the authors for their response. Most of the concerns have been adequately addressed, except for the issue regarding the impacts of sample ordering on FFT. Overall, I will increase the score.

---

> > > ### Author Response · Authors · 2025-12-02
> > > **Reply to Reviewer seS7**
> > >
> > > Thank you! The impacts of sample ordering on FFT is an open challenge, which is not the main concern of this paper. Through the actual experimental results, our method is effective and stable. In the further, we would consider this.

---

### Official Review · Reviewer_acAD · 2025-10-30

**Soundness:** 4
**Presentation:** 3
**Contribution:** 3
**Rating:** 6
**Confidence:** 4

**Summary:**

This paper proposes a novel ARSL-IMVC method for the classic incomplete multi-view clustering task, which aligns the collaborative view recovery and tensorial subspace learning in cross-view complementarity and consistency exploration. It is interesting, which is the inverse idea of classical multi-view clustering. It uses an unknown latent shared representation to reverse infer missing multi-view data and act on the subspace representation learning. Latent representation as a bridge helps to explicitly realize cross-view correlation exploration. The extensive experimental results verify its effectiveness.

**Strengths:**

(1) This paper is clearly motivated and proposes to achieve the "semantic alignment" of view recovery and subspace learning through a shared latent representation in incomplete multi-view clustering, which is innovative.
(2) The logic and writing of the paper are satisfactory.
(3) Experiments on several datasets with multiple missing ratios provide convincing evidence of the clustering performance robustness.

**Weaknesses:**

(1) The choice of latent representation dimensions remains empirical, lacking selection guidance or systematic analysis.
(2) For clustering tasks on large-scale datasets, runtime efficiency is crucial. But the paper does not provide an analysis of runtime, leaving its practical scalability unclear.

**Questions:**

(1) How sensitive is the model to the dimension of the latent representation? Could a too-small latent space limit the recovery ability? Please elaborate and explain further.
(2) The low-rank tensor constraint may lead to over-smoothing of the subspace representations. How does the model maintain clustering discriminability under such constraint?
(3) The proposed model seems to be complex; does it have an advantage in training time compared to existing methods? Please provide relevant comparative experiments.

---

> ### Author Response · Authors · 2025-11-19
> **Reply to Reviewer acAD**
>
> Thank you very much for your suggestions. We now provide further clarification on these questions:
>
> *(1) How sensitive is the model to the dimension of the latent representation? Could a too-small latent space limit the recovery ability? Please elaborate and explain further.*
>
> **Reply:** The latent representation plays an alignment role in the proposed ARSL-IMVC method , so the choice of its dimensions is very important. For datasets with relatively small feature dimensions (e.g., 64 feature dimensions for 100leaves), we selected a smaller $k$. For datasets with thousands of feature dimensions, we generally get the optimal result when $k=20$. Experimental analysis shows that within the commonly used value range {5, 10, 15, $\cdots$, 30, 35}, the model performance remains relatively stable. There are two main reasons for setting a relatively smaller $k$: 1) Cross-view shared feature space is usually low-dimensional; 2) This could relatively reduce computational complexity. Further, the recover ability from relatively low-dimensional latent space could be guaranteed, which is mainly attributed that: 1) The explicit feature reconstruction constraint could facilitate the latent presentation including cross-view complementary information and improve its recover ability; 2) The HSIC regularization term between view-specific information in the CVR module provides more discriminative information, maintaining the view-specific structure. Therefore, an relatively small $k$ does not cause a catastrophic decrease in its recovery capability.
> | Dataset   |   5   |   10  |   15   |   20  |   25   |   30   |   35   |
> |-----------|-------|-------|--------|-------|--------|--------|--------|
> | BBCSport  | 67.46 | 75.18 | 90.99  | 96.51  | 88.40  | 88.05  | 88.24  |
> | BDGP      | 48.84 | 56.07 | 47.32  | 55.28 | 53.32  | 47.12  | 46.92  |
> | yale      | 69.70 | 75.27 | 77.94  | 86.05 | 90.73  | 89.33  | 85.70  |
> | NGs       | 91.40 | 97.80 | 94.80  | 96.20 | 95.60  | 97.80  | 95.60  |
> | 100leaves | 90.36 | 90.86 | 93.00  | 92.50 | 72.25  | 71.31  | 70.50  |
>
>
> *(2) The low-rank tensor constraint may lead to over-smoothing of the subspace representations. How does the model maintain clustering discriminability under such constraint?*
>
> **Reply:** In this study, the proposed ARSL-IMVC learns shared and view-specific subspace representations and stacks them into a unified low-rank tensor. The low-rank tensor constraint essentially aims to explore the high-order cross-view correlations. Different from existing tensor-based multi-view clustering methods, the ARSL-IMVC integrates cross-view shared subspace representation from high-level and view-specific subspace representation from low-level, promoting the cross-view and cross-level semantic interaction. In fact, the ARSL-IMVC enforces a low rank structure on stacked and rotated subspace representations, which could avoid completely consistent cross-view representation (i.e., over-smoothing issue). In addition, the feature recovery reconstruction and the HSIC diversity regularization term also adds complementary information to each view, preserving the diverse subspace representations and maintaining the clustering semantic discriminability.
>
> *(3) The proposed model seems to be complex; does it have an advantage in training time compared to existing methods? Please provide relevant comparative experiments.*
>
> **Reply:** The design model seems to be complex, but each module is necessary for cross-view semantic alignment in view recovery and subspace representation learning. The computational complexity of the proposed method mainly lies in the ADMM optimization algorithm. To further illustrate the actual efficiency of the model, we supplemented the results with running time experiments. The following table compares the running time (seconds/iteration) in NGs, BBCSport, 100leaves, and BDGP datasets. The experiments demonstrate that the actual running costs of our method are at a moderate level. While not optimal in computational efficiency, it achieves a good balance between performance and efficiency. In the further, we would try to reduce the computational complexity by anchor graph learning strategy with small scale, rather than complete $n \times n$ subspace representation graph.
>
>
> | Methods   |   NGs  | BBCSport | 100leaves |   BDGP  |
> |-----------|-------:|---------:|----------:|--------:|
> | BSV       | 0.0146 | 0.8617   | 0.2263    | 1.2435  |
> | Concat    | 0.1168 | 1.7270   | 0.0711    | 0.2440  |
> | DAIMC     | 3.5722 | 9.3949   | 1.3958    | 1.0345  |
> | UEAF      | 0.6108 | 0.5109   | 1.3746    | 2.7080  |
> | IMSC-AGL  | 0.3718 | 0.3429   | 3.6214    | 8.5792  |
> | HCP-IMSC  | 0.3953 | 0.2910   | 3.4393    | 6.8946  |
> | BWIC-TIMC | 0.2921 | 0.2557   | 7.2357    | 8.9066  |
> | RMoGL     | 0.4818 | 0.5331   | 3.4361    | 12.4075 |
> | HCLS-CGL  | 0.2196 | 0.1280   | 1.6277    | 3.2471  |
> | Ours      | 0.5333 | 0.5174   | 2.7311    | 5.8518  |

---

### Meta-Review · Area_Chair_Yn4Q · 2026-01-02

**Summary:**

This paper proposes an “incomplete multi-view clustering” method named ARSL-IMVC, which uses a “shared latent representation” to simultaneously perform missing-view imputation and subspace learning. Because this idea has already been applied in many fields, including medical-image analysis and bioinformatics, so its novelty is limited. Most reviewers’ comments are positive, yet they also raise numerous concerns. The authors have responded to all reviewers, and some of them have indicated that they will raise their scores.

**Reviewer Concerns:**

Including but not limited to “the issue regarding the impacts of sample ordering on FFT”.

**Reviewer Scores:**

Most reviewers will maintain their original scores, while a small number will raise them.

---

### Decision · Program_Chairs · 2026-01-26

Accept (Poster)